

# Characteristics and dynamics of extreme winters in the Barents Sea in a changing climate

Katharina Hartmuth[1], Heini Wernli[1], and Lukas Papritz[1,2]

[1]Institute for Atmospheric and Climate Science, ETH Zurich, Zurich, Switzerland
[2]European Centre for Medium-Range Weather Forecasts, Bonn, Germany

**Correspondence:** Katharina Hartmuth (katharina.hartmuth@env.ethz.ch)

**Abstract.** The Barents Sea is experiencing large trends in sea ice decline and increasing surface temperatures while at the same time, it is a key region of weather variability in the Arctic and therefore predestined for the occurrence of surface weather extremes. In this study, we identify extreme winter seasons in the Barents Sea, based on a multivariate method, as winters with large seasonal-mean anomalies in one or several surface parameters encompassing surface temperature, precipitation, sur-

face heat fluxes and surface net radiation. Using large-ensemble climate model data for historical (S2000) and end-of-century (S2100) projections following a RCP8.5 emission scenario, we find distinct clusters of extreme winters that are characterized by similar combinations of anomalies in these key surface weather parameters. In particular, we find that, during extreme winters, seasonal-mean anomalies in surface temperature are usually spatially extended with a maximum over sea ice in S2000 simulations, which shifts towards the continental land masses in a warmer climate, as the formation of a warm or cold air

reservoir is being hampered by the increasing area of open ocean. Several extreme winters are selected for a detailed investigation of their substructure focusing on the relative importance of anomalies in the occurrence of synoptic-scale weather systems and anomalous surface boundary conditions for the formation of such seasons. Large combined anomalies in the key surface parameters result mainly from the accumulation of recurrent short-term events that are linked to distinct patterns of anomalous frequencies in cyclones, anticyclones and cold air outbreaks. While large seasonal-mean anomalies in surface air temperature

can be linked to large-scale patterns facilitating the horizontal advection of relatively warmer (colder) air, which coincides with a lack (surplus) of cold air outbreaks, precipitation anomalies are characterized by local anomalies in cyclone and anticyclone frequency. Additionally, anomalous surface boundary conditions - that is sea ice concentration and sea surface temperatures - facilitate the formation of persistent anomalous surface conditions or further enhance atmospherically driven anomalies due to anomalous surface heat fluxes. In a warmer climate, we find extreme winters with similar substructures as in S2000. How-

ever, the increasing distance of the Barents Sea to the sea ice edge causes a decreasing magnitude in seasonal-mean anomalies of surface air temperatures and the atmospheric components of the surface energy balance. A decrease in the variability of both sea ice and sea surface temperatures indicates a decreasing importance in anomalous surface boundary conditions for the formation of future extreme winters in the Barents Sea, while the robust link shown for surface weather systems persists in a warmer climate.





## 1 Introduction

Global warming strongly affects the Arctic region, causing a rapid increase in surface temperatures and, at the same time, dramatic reductions in sea ice cover (e.g., Parkinson et al., 1999; Cavalieri and Parkinson, 2012; Serreze and Meier, 2019). Global climate models project continuing large changes in Arctic sea ice extent and surface conditions in the coming century (e.g., Stroeve et al., 2007; Notz and SIMIP Community, 2020). The ongoing warming will lead to drastic reductions in sea

ice cover particularly in autumn and winter, with the prospect of an ice-free Arctic during September within a few decades (Chapman and Walsh, 2007; Vavrus and Holland, 2021). Simulations show regionally and seasonally differing trends in key surface variables, which are likely related to the increasing seasonality in Arctic sea ice cover and associated sea ice variability (e.g., Huang et al., 2017; Mioduszewski et al., 2019).

Some of the largest trends in surface air temperatures and sea ice extent are observed in the Barents Sea (e.g., Parkinson et al., 1999; Cavalieri and Parkinson, 2012; Johannessen et al., 2016; Rantanen et al., 2022), which are further enhanced by an increase in the heat transported by the Atlantic inflow (Årthun et al., 2012; Smedsrud et al., 2022). On top of this trend, large variations in sea ice extent, particularly in winter, result in considerable fluctuations of surface conditions such as surface air temperatures and surface heat fluxes between years, making the Barents Sea a key region of Arctic interannual variabil-

ity (van der Linden et al., 2016; Dörr et al., 2021). Thereby, sea ice variability has been found to be influenced by oceanic heat transport on interannual to multi-decadal timescales (e.g., Johannessen et al., 2016; Reusen et al., 2019), while on daily to weekly timescales, synoptic-scale weather systems are key drivers of variable surface conditions in the Barents Sea (e.g., Woods et al., 2013; Graversen and Burtu, 2016; Messori et al., 2018; Papritz, 2020). Extratropical cyclones link the Barents Sea to the mid-latitudes and facilitate the transport of warm and moist air into the region, while cold and dry polar air is advected

behind their cold front. Enhanced local baroclinicity along the sea ice edge further favors Arctic cyclogenesis in the Barents Sea, making it a hot spot of Arctic cyclone variability in winter (Inoue and Hori, 2011; Madonna et al., 2020; Caratsch, 2022).

In recent decades, many studies have addressed linkages between the Arctic and mid-latitude weather and thereby often identified the Barents Sea as key region for possible teleconnections (e.g., Petoukhov and Semenov, 2010; Inoue et al., 2012;

Jaiser et al., 2012). For example, winter cold extremes in the mid-latitudes have been found to coincide with a reduction in late autumn sea ice in the Barents and Kara Seas (Honda et al., 2009; Cohen et al., 2014). Different pathways have been suggested to explain such a pattern, most of which argue that lower tropospheric warming resulting from anomalous oceanic heat loss causes circulation changes that impact mid-latitude weather such as an intensification of the Siberian High (Honda et al., 2009; Inoue et al., 2012; Outten et al., 2023). Enhanced upward activity in the Barents and Kara Seas has further been associated

with a weakening of the polar vortex and subsequent impact on the North Atlantic oscillation pattern (Nakamura et al., 2016; Kolstad and Screen, 2019; Siew et al., 2020). Although the debate about involved processes and causal relationships is still ongoing, it becomes clear that the variability of atmospheric conditions in the Barents Sea can have far-reaching effects on



mid-latitude weather and extremes thereof.

It is one aim of this study to investigate the relative importance of key surface parameters such as surface air temperature, precipitation and the atmospheric components of the surface energy balance in a changing Arctic climate. The projected ongoing warming and sea ice loss will continue to affect surface conditions in the Barents Sea, for example due to changes in the surface energy balance that result in enhanced oceanic heat loss in winter (e.g., Semmler et al., 2012) or the intensification of the water cycle causing a wettening of the region (Bintanja and Selten, 2014; Ford and Frauenfeld, 2022). At the same time, the

decreasing sea ice variability is expected to further modify both interannual variability and seasonality of surface conditions (Boer, 2009; Bintanja and Selten, 2014; Mioduszewski et al., 2019), while the remaining, thinned sea ice cover becomes even more susceptible, particularly to intense cyclones (Aue et al., 2022). Although changes in the atmospheric circulation are subject to a partially large uncertainty, most climate models project a slight increase in winter cyclone frequency in the Barents Sea (e.g., Orsolini and Sorteberg, 2009; Akperov et al., 2019; Oh et al., 2020). However, a reduction in local baroclinicity following

the strong sea ice retreat might cause a decline in cyclone genesis and intensification in this area, which might result in less intense cyclones (Rinke et al., 2017; Day et al., 2018). Furthermore, the projected northward shift of regions experiencing the most intense cold air outbreaks (CAOs) is associated with the strong sea ice retreat (Kolstad and Bracegirdle, 2008; Zahn and von Storch, 2010).

Due to its large sea ice variability and high storm activity in winter, the Barents Sea is very susceptible to extreme weather events such as unusual warm air advection which can cause significant sea ice melt (e.g., Boisvert et al., 2016; Cullather et al., 2016) and rain-on-snow events (Overland, 2022), as well as intense CAOs from the Arctic interior, which can trigger strong air-sea heat exchanges (Fletcher et al., 2016; Papritz and Spengler, 2017). As such events can have direct and major impacts on the environment and communities in the region of the Barents Sea, they have been the focus of many studies during recent

decades (e.g., Binder et al., 2017; Rinke et al., 2017; Messori et al., 2018; Overland, 2021). The accumulation of extreme weather events over several weeks to months can result in extreme winter seasons. Thereby, the concept of extreme seasons addresses a timescale between short-term weather events on a daily-to-weekly timescale and long-term trends on a climatological timescale.

Recent studies investigating extreme seasons in the Arctic region have been focusing mainly on seasonal Arctic temperature extremes. New approaches include the identification of Arctic extreme seasons based on the combination of several surface parameters as opposed to one particular variable as shown in Hartmuth et al. (2022), hereafter referred to as HA2022. There, we introduced a new multivariate method using Principal Component Analysis (PCA) to determine in an objective way the "unusualness" of a season considering seasonal-mean anomalies in surface air temperature, precipitation, surface heat fluxes

and surface net radiation. By applying this approach to ERA5 reanalysis data we showed that the formation of Arctic extreme seasons is highly variable and strongly determined by both atmospheric variability and surface boundary conditions, the latter particularly in regions with high sea ice variability such as the Barents Sea. While seasonal-mean anomalies in key surface



variables were found to mainly result from an accumulation of anomalies following frequent events of weather systems on a daily to weekly timescale, anomalous surface boundary conditions cause more persistent anomalies throughout a season, which can similarly favor the formation of such conditions. However, a main limitation of HA2022 is the small number of extreme seasons in the ERA5 dataset. Thus, we aim here to provide a statistically robust analysis of the substructure of such seasons.

The dramatic changes in surface and atmospheric conditions in the Barents Sea further suggest that future extreme winters will look differently compared to extreme winters in the present-day climate. Recent studies using climate model simulations showed that Arctic winter temperature extremes become warmer for hot extremes and less severe for cold extremes, whereby cold extremes warm faster than hot extremes (Saha et al., 2006; Kharin et al., 2013; Lo et al., 2023). This is particularly relevant in the Barents Sea, where the increasing distance to the sea ice edge is projected to contribute to a strong reduction in surface air temperature variability and cold extremes are projected to warm dramatically (Saha et al., 2006; Hartmuth et al., 2023). Further, climate models project an increase in frequency and intensity of both winter warm events and precipitation extremes (Saha et al., 2006; Kharin et al., 2013; Graham et al., 2017; Bogerd et al., 2020). Due to the increase in mean temperature, the rainfall ratio will increase strongly in future precipitation, which additionally enhances the probability of severe conditions such as rain-on-snow events (Bintanja et al., 2020; McCrystall et al., 2021).

Many questions remain with regard to the formation and characteristics of extreme winters in the Barents Sea, in particular in a warming climate. For one thing, the relative importance of changes in the seasonal mean vs. interannual variability of surface parameters is still under debate (Overland, 2022; Lo et al., 2023). Further the question arises if future extremes are driven by similar dynamic and thermodynamic processes as present-day extremes and how the drastic change in surface conditions will affect their characteristics. A detailed and systematic analysis of extreme winters in the Barents Sea, as performed in this study, can lead to new insights regarding the processes leading to such seasons and their relation to the general atmospheric circulation. The comparison of extreme winters in the current and future climate will provide a better understanding on how the formation of winter extremes on the seasonal scale and their impact possibly change in a warmer world.

The goal of this paper is to complement the results in HA2022 with a detailed analysis of the characteristics and dynamics of extreme winters in the Barents Sea in both CESM1 present-day (S2000) and end-of-century (S2100) simulations. Thereby, the more than 1000 years of simulations per time period allow for a robust statistical investigation of such extreme winters and the evaluation of future projections. The aim is to address the following research questions:

1. Does the relative importance of selected surface parameters for the interannual variability of winters in the Barents Sea change in a warmer climate?

2. What is the substructure of extreme winters in the Barents Sea and are they related to the unusual occurrence of synoptic-scale weather systems and anomalous boundary conditions?

3. To what extent do these characteristics change in a warmer climate?



This study is organized as follows: The data and methods used are presented in Sect. 2. Results of the PCA analysis are discussed in Sect. 3 with particular emphasis on climate change effects. Subsequently, we present several case studies of extreme winters in the Barents Sea with a focus on their substructure and the role of synoptic-scale weather systems for the
formation of anomalous surface conditions on the seasonal timescale. Results for simulations in S2000 are presented in Sect. 4 and for S2100 in Sect. 5, before we conclude our results in Sect. 6.

## 2 Data and method

### 2.1 CESM1 data

In this study, we assess simulations with the fully-coupled Community Earth System Model version 1 (CESM1; Hurrell et al.,
2013), which are initialized by using restart files from the CESM large ensemble project (CESM-LE; Kay et al., 2015). In addition to the original CESM-LE data consisting of 35 ensemble members, further simulations are initialized as described in Röthlisberger et al. (2021). In total this results in a 105-member ensemble for a historical period (S2000; 1990−2000) and an end-of-century period (S2100; 2091−2100). For S2100, simulations are run under a representative concentration pathway 8.5 (RCP8.5) forcing scenario. At a spatial resolution of approximately $1°$ we use daily and seasonal-mean surface-level fields
of 2 m temperature ($T$), precipitation ($P$; sum of rain and snow), surface sensible ($H_S$) and latent ($H_L$) heat fluxes, surface net shortwave ($R_S$) and longwave ($R_L$) radiation, sea surface temperature (SST) and sea ice concentration (SIC). In addition, the sum of surface heat fluxes and surface net radiation is defined as the surface energy budget and denoted by $E_S$. SST is only defined at grid points over the ocean where $SIC \leq 50\%$. Throughout this study, we denote anomalies of a variable $\chi$, i.e., deviations from climatology, as $\chi^*$, usually referring to seasonal-mean anomalies except if stated otherwise.


For this study, we focus on the winter season (December−February, DJF). Our analyses are performed for the region of the Barents Sea that is already mostly ice-free in S2000. To this end, we apply a threshold for the winter-mean sea ice concentration over all simulated years in S2000 ($SIC_{S2000}$) of 50% and define the area of the Barents Sea where $SIC_{S2000} < 50\%$ as our region of interest, referred to as BS. Irrespective of the changing sea ice coverage, the same region is used for the analysis of
S2100 simulations. When analyzing spatio-temporal averages of the surface parameters, we first take the seasonal mean before calculating a spatial average over the BS region.

To investigate synoptic features such as cyclones and anticyclones, we apply weather system identification schemes as described in Sprenger et al. (2017). Based on 6-hourly sea level pressure (SLP) data, cyclones and anticyclones are defined as
objects that cover the area around a SLP minimum and maximum, respectively, and are thereby delimited by the outermost closed SLP contour. We further define marine CAOs at grid points over the ocean where $SIC \leq 50\%$, at times when the 900 hPa sea-air potential temperature difference ($\theta_{SST}-\theta_{900}$) is larger than $\geq +4\,K$ (Papritz and Spengler, 2017).





Each weather system is identified as an object described by a two-dimensional binary field with a value of 1 at grid points
inside the object, and 0 outside. By time-averaging these binary fields, we calculate fields of mean cyclone frequency ($f_c$),
anticyclone frequency ($f_a$) and CAO frequency ($f_{CAO}$). For example, $f_c = 0.3$ at a given grid point indicates that at 30 % of
all times, a cyclone is present at that grid point. Spatio-temporal averaging over the region BS yields daily or seasonal-mean
cyclone frequencies for this area. Similarly, the winter-mean weather system frequency anomaly is calculated as the deviation
of the spatially averaged winter-mean weather system frequency from the climatology. We further consider relative frequency
anomalies, which for a specific weather system are calculated (in percentage) as:

$$\frac{f_{\text{seas}} - f_{\text{clim}}}{f_{\text{clim}}} * 100 \tag{1}$$

where $f_{\text{seas}}$ and $f_{\text{clim}}$ denote the spatially averaged winter-mean weather system frequencies for the season and the climatology,
respectively.

## 2.2 Definition of extreme seasons

Extreme winters in BS are identified using the PCA-based method introduced in HA2022. The PCA allows to reduce the
dimensionality of a six-dimensional phase space, spanned by the spatially averaged seasonal-mean anomalies of $T$, $P$, $H_S$,
$H_L$, $R_S$, and $R_L$, to two dimensions. Thereby, the resulting first and second principal component (PC1 and PC2) maximize
the explained proportion of the total inter-seasonal variability in the six-dimensional precursor phase space. Due to the com-
bination of several parameters, the interplay between these variables, which is largely affected by the surface type (sea ice vs.
open ocean) is taken into account. This makes the method applicable to different seasons and Arctic sub-regions with varying
climatological sea ice conditions and allows for a comparison of extreme seasons considering the heterogeneity of the Arc-
tic surface. In addition, the multivariate approach enables the investigation of a broader spectrum of extreme seasons, as also
seasons that are extreme only due to the unusual combination of anomalies in several surface parameters are taken into account.

The result of the PCA can be illustrated using a biplot as shown in Fig. 1. Each dot represents one winter season in BS and
the distance of a season to the origin of the PC1−PC2 phase space, the so-called "Mahalanobis distance" ($d_M$), is a measure
for the unusualness of the season (for more details see Sect. 2 in HA2022). Here, the 50 winters with the largest $d_M$ are defined
as extreme winters (dots outside the red circle in Fig. 1). This corresponds to a return period of approximately 40 years, which
has been used as an adequate measure for defining extreme seasons by other studies, e.g., Röthlisberger et al. (2021).

Radial vectors show the relative contribution of the precursor variables to PC1 and PC2, where relatively shorter (longer)
vectors indicate a smaller (larger) contribution of the precursor to the explained variance. The relative position of two vec-
tors indicates the correlation between both corresponding precursors, with uncorrelated precursors resulting in approximately
perpendicular vectors and more strongly correlated precursors resulting in more parallel vectors. The representation of this
correlation in a PCA biplot is, however, dependent on the variance explained by PC1 and PC2 (Gabriel, 1971, 1972). The
relative position of a season with respect to the precursor vectors refers to the contribution of the different surface variables to





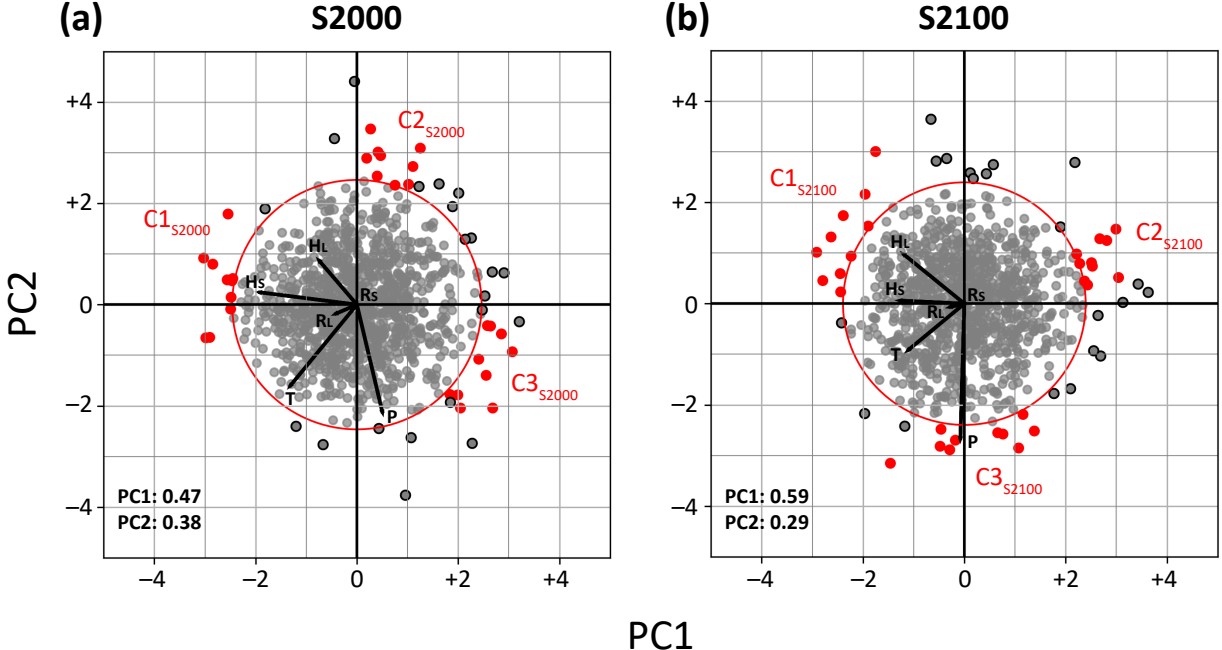

**Figure 1.** PCA biplot for BS in (a) S2000 and (b) S2100 with PC1 and PC2 along the x-axis and y-axis, respectively. Normal seasons are represented by grey dots, extreme seasons are those outside the red circle, which represents the 50th-largest $d_M$ value. Three clusters of extreme seasons in each period are indicated by red dots. Black lines represent the coefficients of the precursor variables. The variance explained by PC1 and PC2, respectively, is given in the lower left corner of each panel.

the $d_M$ value of this season. For example, colored seasons at the top of the biplot shown in Fig. 1a are characterized by negative seasonal-mean anomalies in $T$ and $P$, while colored seasons to the left show positive seasonal-mean anomalies in $H_S$.

## 2.3 Identification of extreme season clusters

195

In order to statistically evaluate extreme winters in BS in current and future climate states, we define clusters of extreme seasons in the respective PCA biplot. Thereby, we aim to contrast collections of similar seasons (i.e., closely spaced in the PCA biplot), whereby the distinct clusters differ strongly in their characteristics (i.e., located in different directions of the PCA biplot). For example, in the S2000 PCA biplot (Fig. 1a) we can already identify several distinct clusters by eye. Now

we introduce a method to identify such clusters, whereby we choose a cluster size of ten seasons to allow for a meaningful statistical comparison between different clusters. First, we determine for each group of ten adjacent extreme seasons the angle segment that is covered by the seasons in this group. In a second step, the three groups with angle segments that do not overlap





and that cover the smallest combined angle segment (red dots in Fig. 1) are chosen for the cluster analysis in Sect. 4 and 5. Due to the distribution of extreme seasons in the PCA biplot for S2100 (Fig. 1b), some compact collections of extreme seasons such as a group of very dry seasons at the top of the biplot are disregarded, as they do contain less than ten seasons, hampering a statistical analysis.

## 3 Interannual variability

### 3.1 S2000

Here we present the results of the PCA analysis of CESM1 simulations in S2000 for the BS region (Fig. 1a) and discuss them against the background of the results shown for a similar region in the ERA5 dataset in HA2022. In S2000, the first two principal components, PC1 and PC2, explain 84.5 % of the total variance of the six-dimensional precursor phase space. This value is slightly lower compared to ERA5 data, where PC1 and PC2 explain even 95 % of the total variance and, thus, capture most of the variance. Winter annual variability in CESM1 is almost in equal parts determined by $T$, $P$ and the surface heat fluxes, whereas $R_L$ is less relevant. $R_S$ is almost irrelevant in winter, as only very little solar radiation reaches the surface during polar night. In the ERA5 dataset, $P$ contributes by far the most to interannual variability in sub-region KBS as defined in HA2022 (which largely overlaps with the BS region used here), while the surface heat fluxes are slightly less relevant than in CESM1.

The relative positions of individual seasons in the biplot (i.e., the PC1 and PC2 scores) are in a similar range compared to the scores in ERA5. Slightly higher scores are reached by some extreme seasons in CESM1, resulting in a larger combined magnitude of the seasonal-mean anomalies of the six precursor variables in the two-dimensional PCA phase space, which we denote by $d_M$ (see also Sect. 2.2). As the number of available seasons is much larger in CESM1 compared to ERA5, it is expected that seasons with larger anomalies can be found in CESM1, particularly as both datasets exhibit a similar variability for most of the precursor variables (see chapter 2 in Hartmuth, 2023).

While in HA2022 we defined extreme seasons in the ERA5 dataset based on a fixed $d_M$ threshold ($d_M \geq 3$), here we choose the 50 seasons exhibiting the largest $d_M$ value as extreme seasons in CESM1 (see Sect. 2.2). This results in a threshold of $d_M = 2.47$ for S2000 simulations. Extreme seasons, shown as dots outside the red contour in the PCA biplots, are not distributed evenly along the edges of the point cloud but instead form clusters at distinct locations in the biplot. It can be expected that seasons within such a cluster have a similar substructure as they exhibit a similar position in the PCA biplot and, thus, combination of seasonal-mean anomalies. For instance, several extreme winters in S2000 are grouped at the top of the biplot (Fig. 1a). As they are located in the opposite direction of the $T$ and $P$ precursor vectors, we can assume that these seasons show negative seasonal-mean anomalies in both $T$ and $P$ and are, thus, particularly cold and dry. We will further analyze the substructure and the large-scale atmospheric pattern linked to this group of extreme winters in Sect. 4.





### 3.2 S2100

We now compare results of the PCA in S2100 simulations to the results for S2000 discussed beforehand. Figure 1b shows the corresponding PCA biplot. In S2100, PC1 and PC2 explain 87.6 % of the total variance for winters in BS, which is a similar value compared to S2000. The threshold for extreme winters in S2000 is $d_M = 2.40$, which implies that future extreme winters show a similar $d_M$ value as extreme winters in a present-day climate.

We first analyze changes in the length of the individual precursor vectors. An increase in precursor length from S2000 to S2100 signalizes a larger projection of the original precursor vector onto the two-dimensional PCA phase space, which indicates a larger contribution of the particular parameter to the interannual variability in S2100 compared to S2000. Vice versa, a decrease in precursor length implies that the associated parameter becomes less relevant in a warmer climate. Changes in the precursor lengths are relatively small for winters in BS compared to other Arctic sub-regions and other seasons (see Fig. 5.4 in 245 Hartmuth, 2023). As the BS region is already largely ice-free in S2000, it experiences less substantial changes in the variability of surface parameters compared to regions that undergo a relatively large sea ice loss, such as for example the High Arctic. Nevertheless, an increase in the contribution of $H_L$ and $P$ to the overall variability is shown in the biplots, while the contribution of $H_S$ and $T$ slightly decreases towards S2100. Seasonal sea ice that is still partially existent in BS in S2000 simulations will disappear in most S2100 simulations resulting in an increased area of open ocean and enhanced evaporation. At the same 250 time, $P$ is projected to increase in the whole Arctic (not shown), causing an increased $P$ variability also in BS. The change towards more homogeneous surface conditions could explain the decreasing importance in $T$ and $H_S$, while the increase in the variability of the other precursor variables could similarly result in such a decrease.

The PCA biplots provide further information about the correlation between the precursor variables, which is indicated by 255 the relative location of the precursor vectors (see Sect. 2.2). In our case, where PC1 and PC2 explain most of the variance, the correlation between two precursors becomes evident from the associated PCA biplot. However, changes in the correlation between precursor variables are comparatively small for BS, which can be linked to the relatively small changes in surface boundary conditions. In particular, the weakly positive correlation between $T$ and $P$ remains almost the same between S2000 and S2100, indicating that the variability in these parameters is governed by similar processes in S2100 as in S2000.


To summarize, small changes in the representation of the key surface variables in the PCA phase space reflect that interannual variability in winter conditions in the Barents Sea will be governed similarly by $T$, $P$, and the surface heat fluxes in a warmer climate with a slightly increasing contribution of $P$ compared to the other surface variables. As the region is already largely ice-free in the present-day climate, changes in sea ice variability are comparatively small and therefore the correlation 265 between the variables remains largely unchanged. We further find no significant changes in future $d_M$ values, which implies that there is no notable increase or decrease in the amplitude of combined seasonal-mean anomalies of key surface variables.





In the following, we will focus on clusters of extreme winters in BS to investigate the characteristics and formation of such seasons in both S2000 and S2100 simulations. However, following the different reference climate states used in the respective

PCAs, the disparate distribution of extreme seasons in the PCA biplot as well as the slightly different combination of precursor contributions, the clusters between both periods are not directly comparable. Therefore, we will analyze both S2000 and S2100 clusters separately and while this evaluation serves less a direct comparison of extreme seasons in present-day and future conditions, we will discuss some general differences between both climate states in Sect. 6.

## 4 Extreme winters in S2000

### 275 4.1 Cluster overview

While the position of clusters of extreme winters in the PCA biplot already gives an idea about the surface parameters that are particularly anomalous in these winters, there is no information from the biplot about the spatial distribution and actual magnitude of these anomalies. Figure 2 shows spatial composites of the seasonal-mean anomalies of $T$ (denoted as $T^*$), $P$ ($P^*$), $E_S$ ($E_S{}^*$) and SST (SST$^*$) for each cluster in S2000.


Winters in cluster 1 (C1$_{S2000}$ in Fig. 1a) are characterized by positive anomalies in surface heat fluxes (particularly $H_S$) and a positive $T^*$. Figure 2a shows an extensive area with positive $T^*$ covering large parts of the Arctic Ocean that is particularly pronounced along the Scandinavian and Russian coastline and over the sea ice covered part of the Barents Sea, where it coincides with a reduction in SIC (yellow lines in Fig. 2a). At the same time, a positive $E_S{}^*$ is particularly pronounced over

the open ocean (Fig. 2g) and spreads far into the Norwegian Sea. The shift of the sea ice edge to the North coincides with a consistently positive seasonal-mean anomaly in SSTs (Fig. 2j). A small negative $P^*$ is shown in BS, which is more evident along the Norwegian coast (Fig. 2d).

Winters in cluster 2 (C2$_{S2000}$ in Fig. 1a) are located opposite to the $T$ and $P$ vectors in the PCA biplot, which lets assume

that these winters are unusually cold and dry. Figure 2b shows indeed a spatially extended negative $T^*$, which is most strongly pronounced in the eastern Barents Sea where the sea ice extends anomalously far South. Interestingly, the spatial extension of this negative $T^*$ is very similar to that of the positive $T^*$ shown for cluster 1 with its maximum over the sea ice covered part of the Barents Sea. The area within BS that experiences anomalous sea ice cover exhibits a pronounced negative $P^*$ (Fig. 2e). In terms of $E_S$, the winter seasons of this cluster exhibit a dipole with a positive $E_S{}^*$ where more sea ice than usual is present and

a negative $E_S{}^*$ over the open ocean (Fig. 2h). While the increased ice cover reduces air-sea interactions, the southward shift of the sea ice edge and, thus, shift in the occurrence of CAOs likely causes anomalous oceanic heat loss over the southwestern part of BS, an area which is climatologically further away from the sea ice edge. The anomalous sea ice cover of this cluster further coincides with a pronounced negative anomaly in SSTs, which extends into the Norwegian Sea (Fig. 2k).



**Figure 2.** Seasonal-mean anomalies of (a-c) $T$ ($T^*$; in K), (d-f) $P$ ($P^*$; in mm day$^{-1}$), (g-i) $E_S$ ($E_S^*$; in W m$^{-2}$), and (j-l) SST (SST$^*$; in K) for extreme winters in cluster 1 (left column), cluster 2 (middle column) and cluster 3 (right column) in S2000. The solid yellow line shows the mean sea ice edge for the respective cluster and the dashed yellow line the climatological sea ice edge. The BS region is shown by the orange contour.



Winters in cluster 3 (C3$_{S2000}$ in Fig. 1a) are located opposite to winters in cluster 1 in the PCA biplot and, thus, include seasons that are characterized by large negative seasonal-mean anomalies in surface heat fluxes. This is illustrated in Fig. 2i, which shows a pronounced negative $E_S{}^*$ in the whole BS region and in particular over the open ocean, where it further coincides with a positive $P^*$ (Fig. 2f). While $T^*$ takes consistently negative values across the region, the center of this negative anomaly is located over the ice-covered part of the eastern Barents Sea and the Kara Sea (Fig. 2c). It is noteworthy that despite

the negative $T^*$, a slightly positive anomaly in seasonal-mean SSTs is shown in region BS and even more pronounced in the Nordic Seas (Fig. 2l). The sea ice edge in these winters is close to climatology.

## 4.2    Substructure

It is one goal of this study to better understand the formation of extreme winters in BS. In this section, we investigate the substructure of such winters by conducting several case studies. This analysis complements the results shown in HA2022 who

performed similar case studies for present-day extreme winters in the ERA5 dataset. One objective here is to investigate the diversity in extreme winters in BS, which has been inhibited using a relatively short record of reanalysis data in HA2022. We further aim to validate the ability of the CESM1 model to represent similar extreme seasons as analyzed by HA2022 in the ERA5 reanalysis. We therefore first investigate the substructure of six extreme winters in BS, whereby always two seasons are part of the same extreme seasons cluster. While winters from the same cluster are expected to show a similar substructure,

large differences between different clusters can be assumed as the clusters feature strongly different seasonal-mean anomalies in the precursor variables, as shown in Sect. 4.1.

### 4.2.1    Cluster 1: DJF 1993/94 (member 008) and DJF 1996/97 (member 090)

We first analyze the substructure of two winters in cluster 1, winters 1993/94 in ensemble member 008 (CS1) and 1996/97 in ensemble member 090 (CS2). The analysis of daily-mean values in the key surface parameters shows almost continuously

positive daily-mean $T^*$, which are correlated with periods of persistent positive anomalies in $E_S$ in both winters (Fig. 3a, b). Notably, daily-mean $T$ values often exceed the range given by one standard deviation, which underlines the unusualness of both winters. The persistence of relatively warm surface temperatures can be associated with persistent positive daily-mean anomalies in SST and a lack of sea ice formation at the same time. In both winters, almost no sea ice has formed until the end of February. Despite the anomaly in open ocean exposure, both winters are characterized by a clear deficit of CAOs (green

heatmap in Fig. 3). As the frequency in passing cyclones is only slightly or not reduced at all compared to climatology (blue heatmap in Fig. 3), such a reduction in $f_{CAO}$ is possibly associated with an anomalous cyclone pathway in these winters which we will investigate in Sect. 4.3.

### 4.2.2    Cluster 2: DJF 1998/99 (member 059) and DJF 1993/94 (member 072)

A detailed analysis of the winters 1998/99 in ensemble member 059 (CS3) and 1993/94 in ensemble member 072 (CS4) shows

that the strongly negative $T^*$ shown for cluster 2 (see Fig. 2b) results from several persistent periods of daily-mean $T$ values



**Figure 3.** Time series of daily-mean (a, f) $T$ (in °C), (b, g) $E_S$ (in W m$^{-2}$), (c, h) SIC, (d, i) $P$ (in mm day$^{-1}$), and (e, j) SST (in °C) averaged for BS in selected extreme winters in S2000 (black lines) for (a) CS1, (b) CS2; (c) CS3, (d) CS4, (e) CS5, and (f) CS6. The climatology is shown by grey lines. Grey shading shows the standard deviation of daily-mean anomalies in all winter seasons of all S2000 members relative to the respective climatology. Blue, orange, and green heatmaps at the bottom of each panel show the daily-mean coverage of BS by cyclones, anticyclones, and CAOs, respectively (the darker the color the higher the coverage). Relative, spatially-averaged frequency anomalies of $f_c$ ($f_{c,clim} = 0.28$), $f_a$ ($f_{a,clim} = 0.04$), and $f_{CAO}$ ($f_{CAO,clim} = 0.42$) are given in %. The horizontal axis indicates days since the start of the season with day 1 corresponding to 01 December.





well below the climatology (Fig. 3c, d). These periods, which are occurring for example between days 7 and 45 in CS3, are further characterized by below-average precipitation and can be associated with a lack of cyclone activity in BS, while in the same period the occurrence of anticyclones likely further enhances surface cooling (orange heatmap in Fig. 3). In addition, large and continuous positive anomalies in SIC and concurrent negative anomalies in SST that already exist at the beginning
of both winters are likely to facilitate the persistent cold conditions. Whereas on average sea ice formation in BS starts at the beginning of December, both winters already feature a sea ice coverage of more than 25 % at this time. Such a reduction in open ocean area is consistent with a reduction of $f_{CAO}$ in both winters, as by default CAOs are not defined over sea ice (see Sect. 2).

### 4.2.3 Cluster 3: DJF 1999/00 (member 008) and DJF 1997/98 (member 052)

While the previously shown winters are partially characterized by pronounced anomalies in surface air temperatures, the winters 1999/00 in member 008 (CS5) and 1997/98 in member 052 (CS6) exhibit relatively small seasonal-mean $T$ anomalies in BS. Interestingly, both winters show persistent negative anomalies in daily-mean $T$ during February, resulting in an overall slightly negative $T^*$ (Fig. 3e, f). These periods coincide with a positive SIC anomaly, which, however, only develops during the second half of both winters. The unusualness of winters in cluster 3 results from a strongly negative $E_S^*$. The timeseries
of both case study winters reveal recurrent events of strongly negative daily $E_S$ anomalies, which can be linked to CAOs that facilitate strong oceanic heat loss in the form of upward $H_S$ and $H_L$. At least in the beginning of both winters, upward surface heat fluxes are further enhanced by a negative anomaly in daily-mean SIC. While both winters are characterized by a relative lack of anticyclones, CS6 also features a reduction in $f_c$.

As expected from their position in the PCA biplot, the three pairs of case studies representing three distinct clusters of extreme winters differ strongly from each other. Winters in cluster 1, which feature a positive $T^*$ and $E_S^*$, are characterized by a lack of sea ice and at the same time a reduced frequency in CAOs. In contrast, winters in cluster 2 feature negative seasonal-mean anomalies in $T$ and $P$. These winters are characterized by an unusually high SIC in BS and concurrent relatively low SST values throughout the season, which facilitates the persistence of cold and dry conditions. Finally, winters in cluster 3 are
characterized by negative seasonal-mean anomalies in $E_S$. The persistent occurrence of CAOs results in several episodes of enhanced heat loss into the atmosphere, causing strong negative daily-mean $E_S$ anomalies and a rapid growth of SIC in BS.

We have shown in this section that anomalies in the occurrence of weather systems facilitate the formation of seasonal anomalies in surface atmospheric conditions. However, we are not yet able to fully understand the large-scale processes as well
as the interplay between weather systems and surface boundary conditions that lead to the formation of such extreme winters. Thus, in the following we analyze the spatial distributions of anomalies in the occurrence of cyclones, anticyclones and CAOs, which will improve our understanding of how weather systems affect the evolution of persistent anomalous surface conditions in BS.





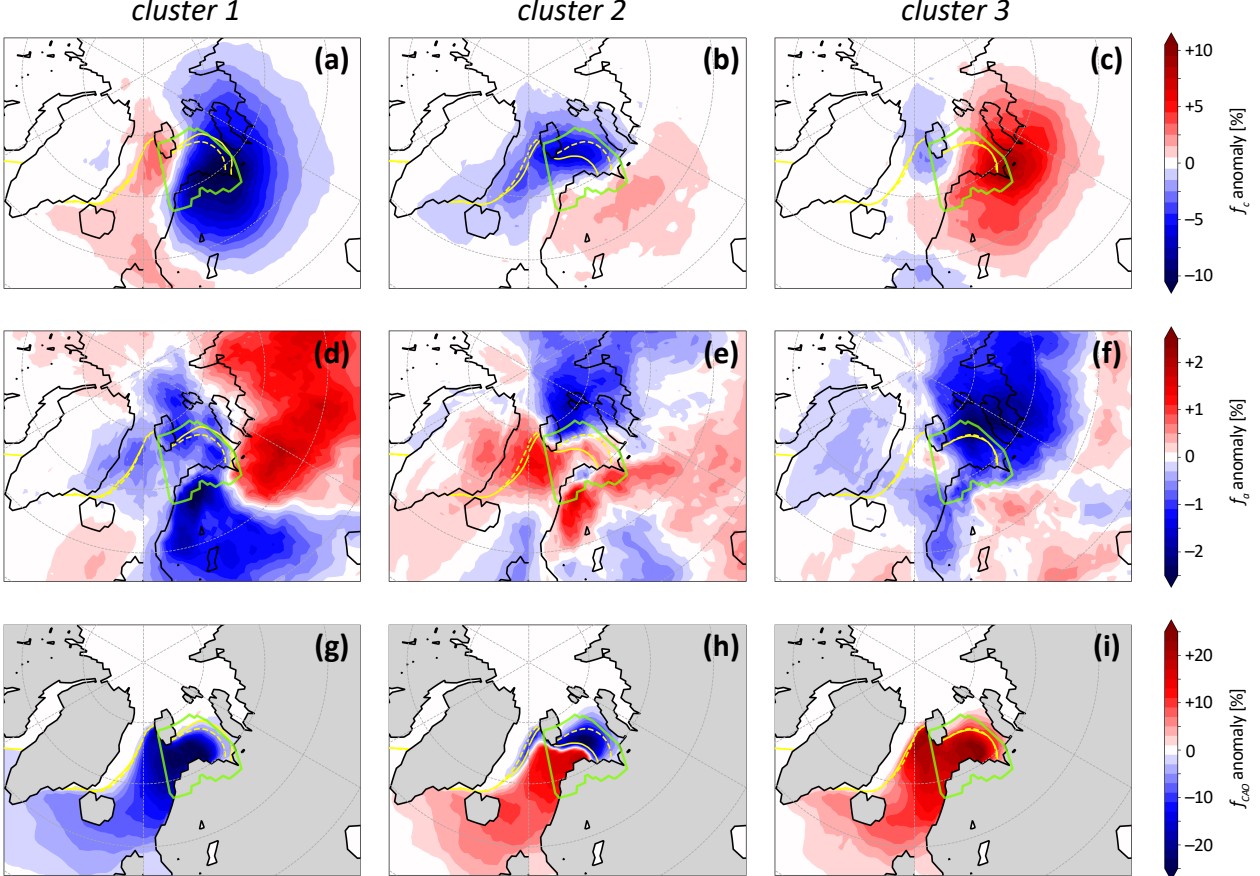

**Figure 4.** Absolute seasonal-mean anomalies in (a-c) cyclone frequency ($f_c{}^*$; in %), (d-f) anticyclone frequency ($f_a{}^*$; in %), and (g-i) CAO frequency ($f_{CAO}{}^*$; in %) for extreme winters in (a, d, g) cluster 1, (b, e, h) cluster 2, and (c, f, i) cluster 3 as described in Sect. 2.3. The solid yellow line depicts the mean sea ice edge for the respective cluster and the dashed yellow line the climatological sea ice edge in S2000 ($SIC_{clim} = 0.5$). The enlarged BS region is shown by the green contour.

## 4.3 Dynamics

To investigate the large-scale dynamics associated with extreme seasons in BS, we analyze anomalies in the occurrence of weather systems that affect this region. For this analysis, we slightly increase the size of region BS to account for the influence of weather systems also if they are not directly colocated with the BS region. In a first step, we examine for each 6-hourly time step during each winter, if the mask of a weather system object (see Sect. 2.1) overlaps with the enlarged BS region. If this is the case, we consider this mask for the calculation of the seasonal-mean weather system frequency anomaly. Note, that such

a frequency anomaly for a specific cluster is calculated as deviation from a climatology, that takes all considered masks into account. As the distribution of this climatology is not necessarily symmetric with respect to the BS region and the number of





values contributing to the climatology per grid point decreases with increasing distance to BS, potential biases following the design of this method will be considered.

We have shown that winters in cluster 1 are mainly characterized by positive anomalies in $T$ and $E_S$ and a concurrent lack of sea ice, while $f_{CAO}$ is reduced. As can be seen from Fig. 4a, a more meridional propagation of cyclones results in a slight surplus of cyclones in the Nordic Seas and the northwestern part of BS and simultaneous lack of cyclones extending from Scandinavia to the Kara Sea which is, however, more pronounced compared to the positive $f_c{}^*$. This is consistent with the results in Hartmuth et al. (2023) who showed a dipole in $f_c{}^*$ for extremely warm winters in this area and at the same time

a local negative $f_c{}^*$ during extremely dry seasons. The composite shown in Fig. 4a can be regarded as combination of these two patterns following the positive $T^*$ and negative $P^*$ featured by winters in cluster 1. Given this pattern it is plausible that during these winters, the advection of cold and dry air in the cold sectors of cyclones is strongly reduced and that most of the time only the warm sector of a cyclone is located in the BS region, causing a net increase in $T$. This could further explain the pronounced lack of CAOs (Fig. 4g) despite a positive SST anomaly in BS (Fig. 2j), which additionally enhances the formation

of a positive $T^*$. Next to a reduced $f_c$ and $f_{CAO}$, winters in cluster 1 also show a weakly negative $f_a{}^*$ (Fig. 4d) in BS. Thereby, a pronounced dipole in $f_a{}^*$ is shown over the continent with a relative lack of anticyclones over Scandinavia and a surplus of anticyclones over Russia which further favors the advection of warm air from the continent (see Fig. 2a).

    In contrast to the warm winters in cluster 1, cluster 2 comprises of particularly cold and dry winters, which presumably

experience a surface preconditioning in the form of a positive SIC anomaly that already exists at the beginning of a winter. Figure 4b shows a negative $f_c{}^*$ for most of BS and in particular for the area with increased SIC, while slightly more cyclones than usual are found over Scandinavia. This pattern is in line with the results found for cold and dry winters in this area in Hartmuth et al. (2023). As cyclones tend to pass to the South of BS rather than directly across the region, the transport of relatively warm and wet air is confined to these areas, while BS is more strongly affected by the advection of cold and dry air

in the wake of these cyclones. At the same time, this pattern favors the advection of air from the cold air reservoir over the continent (see Fig. 2b). This cluster further features a dipole with a positive $f_a{}^*$ over Scandinavia and the Nordic Seas and vice versa a negative $f_a{}^*$ towards the North of BS (Fig. 4e). This pattern further favors the advection of relatively cold and dry air from the High Arctic towards the open ocean. The combination of an enhanced advection of cold and dry air from the regions with exceptionally cold air over the sea ice and the continent as well as a shifted sea ice edge results in a positive $f_{CAO}{}^*$ over

the open ocean in BS (Fig. 4h). Simultaneously, a concurrent reduction in CAOs is found in the area of extended sea ice cover.

    Winters of cluster 3 exhibit recurrent events of strong oceanic heat loss resulting in an overall negative seasonal $E_S{}^*$, which can be linked to the frequent occurrence of CAOs as shown in Sect. 4.2.3. Figure 4i shows a pronounced positive $f_{CAO}{}^*$ over BS and the Nordic Seas. As no significant anomaly in SIC can be found for these winters, an eastward shift in the occurrence

of cyclones as shown in Fig. 4c is most likely causing this dipole in $f_{CAO}{}^*$. The increased occurrence of cyclones towards the East of BS favors the advection of cold and dry air from the ice-covered northern part of the Barents Sea (see $T$ anomaly





in Fig. 2c) and the Kara Sea, which then results in enhanced surface heat fluxes into the atmosphere. At the same time, this pronounced positive $f_c{}^*$ facilitates the overall positive $P^*$ of seasons in this cluster. The relative absence of anticyclones over the eastern Barents Sea, the Kara Sea and along the Siberian coast indicates that the negative $T^*$ in this region is mainly driven
by the shift in cyclone-related advection of cold and dry air as opposed to radiative cooling within persistent anticyclones.

When comparing anomalies in weather system frequencies for the different clusters, it becomes apparent that each cluster is characterized by distinct patterns of such anomalies, which can be related to typical patterns associated with exceptionally large seasonal-mean anomalies in $T$ and $P$ (Hartmuth et al., 2023). In addition, distinct anomalies in $f_{CAO}$ are found for
each cluster and can be linked to both the anomalous frequencies and pathways of cyclones, which affect the advection of cold and dry air from the Arctic interior and Eurasian continent towards BS, and shifts in the sea ice edge. This combination of anomalous circulation patterns and anomalous surface boundary conditions can cause the formation of a strong dipole in $f_{CAO}{}^*$ such as for cluster 2, which further results in a dipole of surface heat flux anomalies (see Fig. 2h), but only a small spatially averaged anomaly. In contrast, for winters in cluster 3, which show on-average sea ice extent, the anomalies in $f_{CAO}$
are mainly caused by anomalies in the atmospheric circulation and cause an overall increase in surface heat fluxes, resulting in a distinct positive $E_S{}^*$. Note that the spatial extension of the CAO anomalies towards lower latitudes is partially affected by the design of the method (see Sect. 2.3) as large coherent masks of CAOs can result if distinct CAO events in the BS region and along the Greenland coast occur at the same time.

After evaluating the processes leading to the formation of extreme winters in BS in the present-day climate, we now repeat this analysis for future winters in BS. In the following section, we again compare three different clusters of extreme winter seasons based on the PCA biplot for S2100 (see Fig. 1b). As mentioned beforehand, due to the design of our method we cannot provide a direct comparison of the same type of extreme winters in both S2000 and S2100. Instead, this analysis aims to shed light on the question how future extreme winters in BS are characterized, how features of their substructure may differ from
present-day extreme winters, and whether the relative importance of weather systems vs. surface boundary conditions for the formation of future extreme winters in BS is expected to change.

## 5   Extreme winters in S2100

### 5.1   Cluster overview

Extreme winters in cluster 1 (C1$_{S2100}$ in Fig. 1b) feature positive seasonal-mean anomalies in the surface heat fluxes and in
particular in $H_L$ (Fig. 1b). Figure 5g shows that a pronounced positive $E_S{}^*$ in BS extends into the Kara and Nordic Seas for these winters. This anomaly coincides with a slightly positive $T^*$, which is, however, more pronounced in the region of the Kara Sea (Fig. 5a). At the same time, a negative $P^*$ occurs, in particular in the southern part of BS, close to the Norwegian coast (Fig. 5d). In terms of SSTs, winters in cluster 1 show no particularly pronounced anomalies (Fig. 5j).





**Figure 5.** Same as Fig. 2, but for extreme winter clusters in S2100.



Cluster 2 (C2$_{S2100}$ in Fig. 1b) includes several unusually cold winter seasons, as can be concluded from their position op-
posite to the $T$ precursor vector in the PCA biplot (see Fig. 1b). Analysing the spatial distribution of this negative $T^*$ shows
that it is most pronounced over the continental land masses of Scandinavia and Russia, while the BS region is only located at
the edge of this anomaly (Fig. 5b). The cold surface air temperatures correlate with SSTs being below climatology in both the
Barents and Kara Seas (Fig. 5k). Further, winters in this cluster feature spatially coherent weakly negative anomalies in both $P$
and $E_S$ (Fig. 5e, h).

Winters in cluster 3 (C3$_{S2100}$ in Fig. 1b) feature particularly wet seasons. Figure 5f shows that the positive $P^*$ is especially
pronounced along the Norwegian coast and in the western part of BS. While these winters show on average only a small
positive $T^*$ in BS, a much larger positive $T^*$ occurs over Scandinavia and Russia (Fig. 5c). This anomaly is possibly linked to
a positive anomaly in SSTs, which is particularly pronounced close to the coast and extends into the Nordic, Barents and Kara
Seas (Fig. 5l). In terms of $E_S$, winters in cluster 3 feature on average only a weakly positive anomaly (Fig. 5i).

## 5.2 Substructure

### 5.2.1 Cluster 1: DJF 2094/95 (member 052) and DJF 2097/98 (member 071)

Winters 2094/95 in ensemble member 052 (CS7) and 2097/98 in ensemble member 071 (CS8) show almost continuous and
correlated positive anomalies in daily-mean $T$ and $E_S$ (Fig. 6a, b). In both winters, a notable lack of CAOs is shown which
likely enables the persistence of warm temperatures at the surface. Additionally, in CS8 the presence of a surface anticyclone
is associated with positive daily-mean $T^*$ and $E_S{}^*$ as well as a lack of $P$ (Fig. 6, orange heatmap). Interestingly, both seasons
feature slightly colder SST values than usual, which indicates that there is no additional contribution to the maintenance of the
warm surface air temperatures by a relatively warm ocean following enhanced SSTs, but that instead the anomalies in both
$T$ and $E_S$ are mainly caused by anomalies in the atmospheric circulation. A surface preconditioning in terms of anomalous
sea ice coverage at the beginning of a winter season as shown for S2000 simulations is almost impossible in S2100 due to the
general absence of sea ice in the BS region.

### 5.2.2 Cluster 2: DJF 2091/92 (member 033) and DJF 2092/93 (member 068)

The two chosen case studies, winter 2091/92 in ensemble member 033 (CS9) and winter 2092/93 in ensemble member 068
(CS10), show prolonged periods of strongly negative daily-mean $T^*$ (Fig. 6c, d). Especially CS9 features several periods with
$T$ being well below the sigma range. In both winters, the cold episodes can be linked to recurrent CAO events and are possibly
maintained by below-average SST values. The frequent CAOs further coincide with episodes of enhanced surface heat fluxes
into the atmosphere during both winters, which result in strongly negative daily-mean $E_S{}^*$. In both winters, the occurrence of
anticyclones, often during periods with reduced $f_c$, can be linked to periods with negative daily-mean $P^*$. While $f_a$ and $f_{CAO}$
are enhanced compared to climatology, particularly DJF 2092/93 features a relative lack in $f_c$.



**Figure 6.** Same as Fig. 3, but for extreme winter case studies in S2100 (a) CS7, (b) CS8, (c) CS9, (d) CS10, (e) CS11, and (f) CS12. Relative seasonally-integrated frequency anomalies of $f_c$ ($f_{c,clim} = 0.31$), $f_a$ ($f_{a,clim} = 0.03$), and $f_{CAO}$ ($f_{CAO,clim} = 0.31$) are given in percentages.



### 5.2.3 Cluster 3: DJF 2091/92 (member 004) and DJF 2099/00 (member 019)

The analysis of the winters 2091/02 in member 004 (CS11) and 2099/00 in member 019 (CS12) shows several precipitation events throughout these winters, which are further characterized by positive anomalies in both $f_c$ and $f_{CAO}$ (Fig. 6e, f). Apart from short episodes where the BS region is affected by a surface anticyclone, daily-mean $P$ values are almost always above cli-

matology in both winters and are frequently associated with the passage of a cyclone. Several episodes of negative daily-mean $E_S{}^*$ are linked to CAOs, which typically occur with a lag of a few days relative to the passage of a cyclone. This indicates that the respective CAO events are associated with the advection of cold and dry air in the wake of a cyclone.

Similar to S2000, the three clusters show very different characteristics, whereby winters within the same cluster exhibit

strong similarities with regard to their substructure, anomalies in weather system frequencies and surface boundary conditions. Winters in cluster 1 feature persistent positive anomalies in $T$ and $E_S$, which can be linked to the absence of CAO events during most of the season. In contrast, winters in cluster 2 are characterized by prolonged periods where cold conditions and anomalous heat fluxes into the atmosphere prevail, which are associated with recurrent CAOs and the presence of anticyclones in the BS region. Several CAO episodes are similarly characteristic for extremely wet winters in cluster 3, however, these

seasons feature an increased cyclone occurrence in BS, which results in negative surface heat flux anomalies and at the same time frequent precipitation episodes.

Although the case studies of S2000 and S2100 cannot be directly compared, it becomes immediately apparent that the climatological conditions in BS change strongly between S2000 and S2100. In S2100, the region becomes ice-free, with SSTs

reaching values between $+6\,°\mathrm{C}$ and $+8\,°\mathrm{C}$ on average during winter. This is in strong contrast to S2000, when SIC starts to increase from the beginning of December and on average 20 % of BS is covered by sea ice by the end of February (e.g., Fig. 3a, third panel). The absence of sea ice in S2100 is associated with a reduced variability in both $T$ and $E_S$, which becomes visible in the reduced sigma range of both variables in Figs. 3 and 6. In general, smaller daily-mean anomalies are found for all parameters except $P$ in S2100 compared to anomalies in S2000. However, the anomalies are still comparable relative to

the respective climatology, owing to the reduced variability in many surface parameters in BS in a warmer climate. Another interesting result is an overall reduction in the correlation between daily-mean $T$ and $E_S$ values in S2100 (not shown), which is probably linked to the stronger reduction in the variability of $T$ ($-66\,\%$) as compared to $E_S$ ($-37\,\%$). Next to this reduction in the amplitude and correlation of daily-mean anomalies in $T$ and $E_S$, we also show similarities in the substructure of winters with similar seasonal-mean anomalies in both S2000 and S2100. For example, negative seasonal-mean anomalies in $E_S$ as

shown for cluster 3 in S2000 and clusters 2 and 3 in S2100 are found to be linked to an enhanced occurrence of CAO events. Similarly, persistent warm conditions such as in cluster 1 in both periods feature a pronounced negative $f_{CAO}$ anomaly.



## 5.3 Dynamics

We now assess the dynamics of extreme winters in S2100 by evaluating anomalies in $f_c$, $f_a$, and $f_{CAO}$, which, again, show pronounced patterns for the distinct clusters. Extremely warm winters in cluster 1 are characterized by a particular lack of
cyclones along the Scandinavian coastline and in the eastern Barents Sea as well as a small positive $f_c^*$ along the Fram Strait (Fig. 7a), a pattern which we have also seen for similar seasons in cluster 1 in S2000 (Fig. 4a). While this pattern favors the advection of relatively warmer air from the North Atlantic, we assume that a large part of the positive $T^*$ in BS results from the absence of cold air advection into this area. This is supported by the strong negative $f_{CAO}^*$ during these winters (Fig. 7g), which is likely linked to the lack of cyclones advecting cold and dry air in their cold sector, facilitating the formation of both
a positive $T^*$ and $E_S^*$ (see Fig. 5a). In addition, we find a dipole in $f_a^*$ with increased anticyclone activity over the Arctic Ocean and a relative lack of anticyclones in a latitudinal band around BS (Fig. 7d).

The cold and dry winters in cluster 2 experience a general lack in both cyclones and anticyclones in BS as cyclones seem to move more zonally to the South of the region (Fig. 7b, e). Thus, these cyclones advect warm and moist air further south, while
facilitating the transport of cold and dry continental air (Fig. 5b) into the BS region. This matches the positive $f_{CAO}^*$ (Fig. 7h), which further favors the persistence of cold and dry conditions. Interestingly, although both case study winters from this cluster show a positive $f_a^*$, which might further facilitate the formation and/or persistence of cold and dry air at the surface due to enhanced radiative cooling, on average the winters of cluster 2 show a weakly negative $f_a^*$ in BS (Fig. 7e). This indicates, that the strongly negative $T^*$ and $P^*$ of these winters are mainly caused by an increase in the frequency of CAO events associated
with a shift in cyclone occurrence as opposed to enhanced radiative cooling in surface anticyclones.

A local surplus in cyclones and simultaneous lack of anticyclones is characteristic for wet winters in both present-day and future simulations (Hartmuth et al., 2023). Here, we show this characteristic also for extremely wet winters in cluster 3 (Fig. 7c, f). A positive $f_{CAO}^*$ with its maximum over the Nordic Seas reflects the enhanced cold air advection associated with
the surplus of cyclones passing through BS in these winters (Fig. 7i).

To summarize, similar to the analyses of S2000 simulations, S2100 extreme winters in BS show pronounced anomalies in the occurrence of synoptic weather systems. In particular, the similar importance of a locally enhanced occurrence of cyclones for extremely wet winters and a decreased occurrence for dry winters in both climate states is in line with our findings in
Hartmuth et al. (2023), where we concluded that the large-scale patterns determining seasonal $T$ and $P$ extremes are largely unaffected by global warming. However, due to the increasing distance of the region to the sea ice edge in a warmer climate, $f_{CAO}$ anomalies are mainly associated with anomalies in $f_c$ and, thus, anomalies in the atmospheric circulation, and less with anomalies in sea ice extent, which reflect anomalous surface boundary conditions.





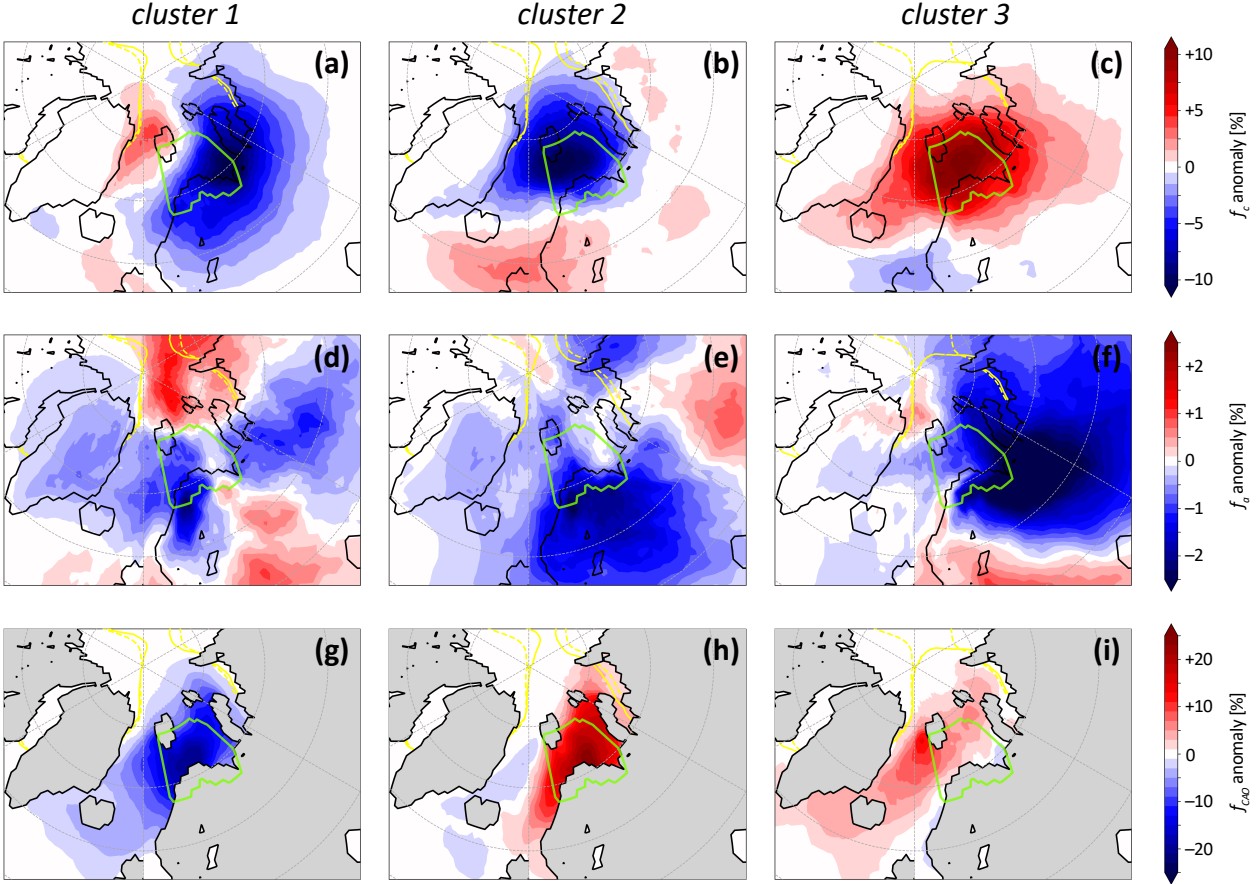

**Figure 7.** Same as Fig. 4, but for extreme winter clusters in S2100 shown in Fig. 1b.

## 6    Discussion and Conclusions

In this study, we investigate the characteristics, substructure and dynamics of extreme winters in the Barents Sea in a chang-
ing climate. Our results extend the analysis of present-day Arctic variability and extreme seasons in the ERA5 reanalysis in
Hartmuth et al. (2022, HA2022) by (1) a statistical analysis of extreme winters in the western Barents Sea (BS) and (2) the
evaluation of future projections using CESM1 simulations. By applying a multivariate approach to identify extreme winter sea-
sons based on the combination of several surface parameters including surface air temperature ($T$), precipitation ($P$), surface
heat fluxes and surface net radiation (which combined yield the surface energy budget $E_S$), we find that in CESM1 individual
parameters contribute similarly to the interannual variability of winters in BS as shown for the ERA5 dataset in HA2022.
Interannual variability in present-day surface conditions is governed to similar parts by the variability in $T$, $P$, and surface heat
fluxes. Changes in the contribution of the individual parameters in a warmer climate are relatively small, which we attribute





to the fact that our region of interest is already almost ice-free in the present-day climate, resulting in comparatively small

changes in sea ice variability.

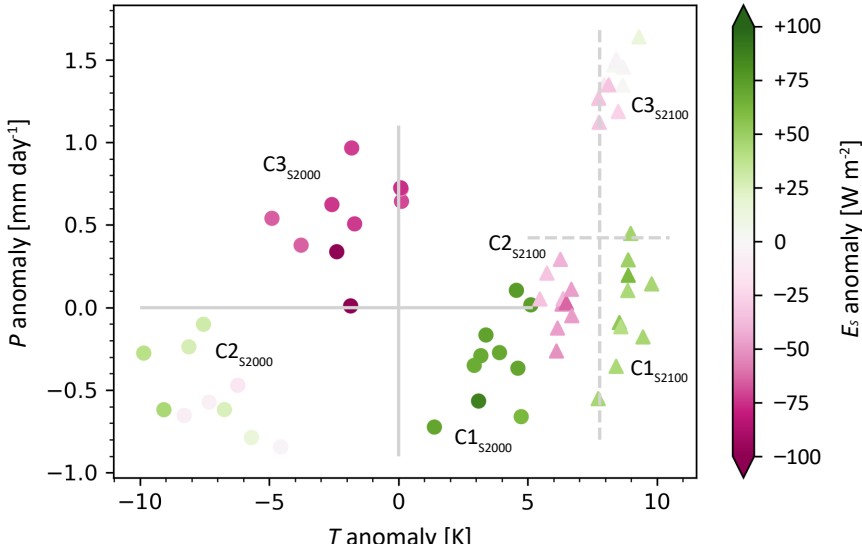

**Figure 8.** Seasonal-mean anomalies with respect to S2000 of $T$ (in K) along the x-axis and $P$ (in mm day$^{-1}$) along the y-axis for extreme winter clusters in S2000 (dots) and S2100 (triangles). Grey lines show climatological mean $T$ and $P$ values in this relative phase space for S2000 (solid) and S2100 (dashed). Markers are colored by seasonal-mean anomalies of $E_S$ (in W m$^{-2}$). Note that for $E_S$, shown anomalies are relative to the respective climatology.

Extreme winters are identified as winters with large combined seasonal-mean anomalies in the above mentioned key surface parameters, whereby these anomalies exhibit similar magnitudes in both ERA5 and CESM1, demonstrating the applicability of this new approach to different datasets. Here, the large amount of available seasons in the CESM1 large ensemble provides us with distinct clusters of extreme winters in BS. While winters within the same cluster feature very similar characteristics such

as for example a positive seasonal-mean $T$ anomaly, a large variety is found between different clusters as various combinations of unusual seasonal anomalies result from our multivariate approach, as shown in Fig. 8. Thereby, the spatial extension of these anomalies is not restricted to the BS region. In particular, seasonal-mean $T$ anomalies usually show a large spatial extent with their maximum located outside BS over sea ice or land areas, emphasizing the role of the ocean as a buffer preventing the formation of large seasonal-mean $T$ anomalies in BS. In a warming climate, we show a shift of the maximum $T$ anomaly from

sea ice covered areas such as the northeastern part of the Barents Sea towards the continental land masses. We conclude that such a shift results from the strongly increased distance of BS to the sea ice edge in S2100 simulations leaving the adjacent land mass as key region to form a warm or cold air reservoir. The projected sea ice decline further results in a decreasing variability of future $T$ and $E_S$, causing a further reduction in the magnitude of seasonal-mean anomalies in future extreme winters (see reduced spread along the x-axis and reduced $E_S$ anomalies of future extreme winters in Fig. 8). Figure 8 puts the





extreme seasons approach into a climate change perspective as it illustrates changes in clusters of extreme winters against the background of an increasing winter-mean $T$ (+7.8 K) and $P$ (+0.42 mm day$^{-1}$) in BS. For example, certain clusters still show similar $T$ and $P$ conditions in both climate states (see C1$_{S2000}$ and C2$_{S2100}$ in Fig. 8), however, the distinct clusters feature completely different prevalent circulation patterns and surface boundary conditions, which results in opposite signs of their seasonal-mean $E_S$ anomaly and highlights the impact of a warming climate on the formation of extreme winters in BS.


It is one goal of this study to investigate the substructure of extreme winters in BS and how it is related to the unusual occurrence of weather systems as opposed to unusual surface boundary conditions. Several case studies of extreme winters show that a variety of involved atmospheric processes causes the substructure of such seasons to be highly variable. We find that large combined seasonal-mean anomalies in surface parameters mainly result from an accumulation of recurrent short-term
anomalous events that can often be related to the anomalous occurrence frequency of weather systems. For example, frequent cold air outbreaks (CAOs) are found to cause periods of strongly enhanced oceanic heat loss, resulting in a large negative seasonal-mean $E_S$ anomaly, while vice versa a reduction in CAO events throughout a winter can result in a positive seasonal-mean $T$ and $E_S$ anomaly. Similarly, wet winters are associated with frequent precipitation events linked to enhanced cyclone frequency. In a warmer climate, a strongly reduced $T$ and $E_S$ variability related to the sea ice retreat cause drastically reduced
daily-mean anomalies throughout such seasons. However, relative anomalies in both key surface parameters and weather system frequencies are comparable for seasons with a similar location in the respective PCA biplot.

To assess the dynamic processes behind the formation of extreme winters in BS, we analyzed the spatial extent of anomalies in the frequency of synoptic features, namely cyclones, anticyclones and CAOs in both climate states. Clusters of extreme winters in BS are characterized by distinct patterns of anomalies in synoptic weather systems, whereby in particular the colocated
presence (absence) of cyclones favors the formation of exceptionally wet (dry) seasons, while $T$ extremes feature patterns that favor anomalous horizontal transport of warm and cold air, respectively, towards BS. These results are in line with our findings in Hartmuth et al. (2023) and confirm the relevance of large-scale atmospheric circulation anomalies associated with anomalous surface conditions. Despite the strong sea ice retreat and projected changes in the frequency of synoptic weather
systems, anomalies in the same surface parameter can be linked to similar patterns of weather system frequency anomalies in a warmer climate, which indicates that the atmospheric processes causing the formation of extreme winter seasons in BS remain similar. Furthermore, we find an increasing importance of the absence of cold air masses for the formation of extremely warm winters in S2100, which might be related to a less effective advection of relatively warmer air into the region due to a decrease in the climatological meridional $T$ gradient and, thus, Arctic $T$ variability (e.g., Screen, 2014; Reusen et al., 2019).


In addition to anomalous weather system frequencies, anomalies in surface boundary conditions, i.e., deviations from the climatological sea ice concentration (SIC) and/or sea surface temperatures (SSTs), can either enhance existing anomalous surface conditions driven by circulation anomalies or facilitate the formation thereof. In the latter case, a pronounced SIC and/or SST anomaly that already exists at the beginning of a winter season and persists throughout the entire winter is denoted as a





surface preconditioning, as such anomalies are found to substantially affect the formation of large combined seasonal-mean anomalies due to the strong linkages between the key surface parameters and sea ice. On one hand, we find extreme winters where anomalous sea ice conditions occur as a result of anomalous surface conditions and, for example, a positive SIC anomaly forms as reaction to recurrent CAO events, causing persistent negative $T$ anomalies and enhanced oceanic heat loss. On the other hand, if the same anomalous oceanic heat loss is caused by a surface preconditioning, it is a negative SIC anomaly that

facilitates such enhanced air-sea interaction. Thus, we find that anomalous surface conditions can be both driven by the atmosphere and by the surface.

Due to the proximity of the BS region to the sea ice edge in S2000, on almost half of the days in DJF a CAO event is detected. We find that extreme winters are characterized by pronounced anomalies in CAO frequency, whereby such anomalies

result from the anomalous advection of air masses, anomalous surface boundary conditions, or a combination thereof in the present-day climate. In a warmer climate with an increased distance to the sea ice edge, anomalies in CAO frequency are mainly attributed to anomalous advection of cold and dry air from surrounding land masses.

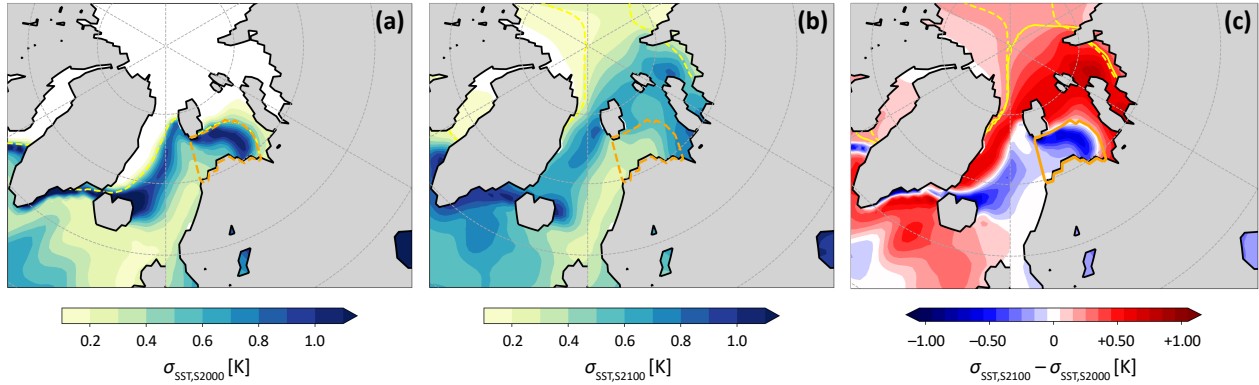

**Figure 9.** SST standard deviation (in K) in (a) S2000 ($\sigma_{\mathrm{SST,S2000}}$) and (b) S2100 ($\sigma_{\mathrm{SST,S2100}}$). Panel (c) shows the difference between S2000 and S2100 ($\sigma_{\mathrm{SST,S2100}} - \sigma_{\mathrm{SST,S2000}}$). The climatological sea ice edge is shown as solid yellow line for S2000 and as dashed yellow line for S2100. Region BS is marked with orange line.

A further objective of this study is to investigate the relative importance of anomalous weather system frequencies as op-

posed to anomalous surface boundary conditions in a warming climate. We find that following the strong reduction in SIC variability in a warmer climate in BS during this century, the relevance of surface boundary conditions decreases while the anomalous occurrence of weather systems remains an essential driver of extreme seasons. This is in contrast to large parts of the remaining Arctic such as for example the Arctic Ocean, where an increase in winter SIC variability conversely results in an increasing relevance of surface boundary conditions for the formation of large seasonal anomalies (Hartmuth, 2023). In



addition, SST variability, which is largest along the marginal ice zone (MIZ), is projected to decrease in BS as well, following
a northward shift in the MIZ as shown in Fig. 9. We conclude that, if surface boundary conditions facilitate the formation of
extreme winters in a warmer climate, this is mainly caused by anomalous SSTs, but in general we can expect a decreasing
importance of anomalous boundary conditions compared to anomalous atmospheric circulation in S2100 compared to S2000.

To summarize, we find that the formation of extreme winters in the Barents Sea is highly variable and strongly determined by
both atmospheric variability and surface boundary conditions following a large sea ice variability in the present-day climate.
We show that large seasonal anomalies in surface parameters in winter can be linked to distinct patterns in weather system
frequencies, which persist in a warming climate. At the same time, the increasing distance to the sea ice edge reduces the
relevance of anomalous surface boundary conditions for the formation of such seasons.


    In this study, we focus on a distinct Arctic region and on winter seasons, and further confine our analyses to surface levels.
A more comprehensive evaluation of extreme seasons in a warming Arctic could involve a comparison of distinct regions with
varying changes in SIC variability, an investigation of different seasons and evaluation of upper-level processes which have
been shown to strongly impact the surface such as upper-level blocking, polar vortex changes or sudden stratospheric warming
events (e.g., Hartmann, 1981; Smith et al., 2018; Domeisen and Butler, 2020). Further, a multi-model setup that applies a more
up-to-date emission scenario will provide a more accurate prediction of the temporal development of Arctic climate change
within the 21st century and allow for a more robust analysis of changes in Arctic variability and extreme seasons. Finally,
when analysing weather systems in the Arctic region, we mainly focus on weather system frequency which has been shown to
be a good metric, for example for the impact of cyclones in the Arctic (e.g., Messori et al., 2018; Papritz, 2020). The analysis
of how other weather system characteristics, such as their intensity, area size or persistence, and changes thereof contribute
to the variability and formation of anomalies in key surface parameters will improve our understanding of the role of weather
systems in driving extreme surface conditions in Arctic regions further. Similarly, an improved knowledge of local interactions
between sea ice and weather systems (Simmonds and Keay, 2009; Ding et al., 2017; Valkonen et al., 2021) will be key for the
assessment of the driving mechanisms behind Arctic extreme seasons.

*Code and data availability.*    The original CESM-LE data (Kay et al., 2015) is available from NCAR's Climate Data Gateway at
https://www.cesm.ucar.edu/community-projects/lens/data-sets. Processed CESM model data including weather system data is openly avail-
able via the ETH Research Collection (https://www.research-collection.ethz.ch/handle/20.500.11850/586054). Scripts used to produce the
analyses and figures in this study are available on request from the authors.



*Author contributions.* KH performed the analyses, produced all figures and wrote the initial draft of the manuscript. KH and HW conceived

the design of the study and the analyses. All authors contributed to the understanding and interpretation of the results and helped to improve the manuscript.

*Competing interests.* Some authors are members of the editorial board of *Weather and Climate Dynamics*. The authors declare that they have no other competing interests.

*Disclaimer.* This study is based on the Doctoral Thesis of the first author, which has been published via the ETH Research Collection

(https://doi.org/10.3929/ethz-b-000637081).

*Acknowledgements.* The authors acknowledge Gary Strand and Clara Deser (both NCAR) for providing the CESM-LE restart files and are thankful to Urs Beyerle (ETH Zurich) for performing the CESM1 simulations. They further thank Michael Sprenger for calculating the weather system data and Hanin Binder, Maxi Boettcher, Mauro Hermann, Matthias Röthlisberger (all ETH Zurich) and Camille Li (University of Bergen) for constructive feedback and helpful discussions. KH acknowledges funding by the European Research Council 485

(ERC) under the European Union's Horizon 2020 research and innovation programme (project INTEXseas, grant agreement no. 787652).



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
