# Peer review of "Characteristics and dynamics of extreme winters in the Barents Sea in a changing climate"

_EGUsphere, 2024_

## Referee Comment (RC3)

**Review of the manuscript**
**Characteristics and dynamics of extreme winters in the Barents Sea in a changing climate**
**By Katharina Hartmuth, Heini Wernli, and Lukas Papritz (WCD-2024-878)**

The aim of the manuscript is to provide a comprehensive analysis of the characteristics and dynamics of extreme winters over the Barents Sea (BS) for both present-day and future conditions, based on large ensembles of climate simulations with the model CESM1. In particular, is study aims to identify the most crucial surface parameters for characterizing extreme winters over the BS, to distinguish different classes (clusters) of extreme winters over the BS and to examine their sub-seasonal evolution and the role of synoptic-scale systems (cyclones, anticyclones, cold air outbreaks (CAO) in the development of these clusters of extreme seasons. Finally, the impact of climate change on these characteristics is investigated. The authors concluded that in the future, anomalous atmospheric circulation will play a more important role than anomalous boundary conditions in the formation of extreme winters.

Given that the BS is both, a hot spot of the current Arctic climate change, exhibiting the strongest temperature amplification and sea ice retreat, and an important region for initiating processes underlying Arctic-midlatitude linkages, particularly the stratospheric pathway in winter, the topic of this study is timely and relevant.

The manuscript forms part of a series of papers on the topic of Arctic extreme seasons (Hartmuth et al., WCD 2022, Hartmuth et al., GRL, 2023). It extends those analyses with an examination of future changes of the characteristics of extreme winter seasons over the BS.

The applied methods comprise a PCA analysis and a cluster analysis, which are not commonly employed in climate studies. In my view, this combination is well suited to the aims of this study and I find it highly intriguing. Many details of the methods and the data-preprocessing can be found in the aforementioned previous papers, while this manuscript provides only a brief overview. This approach may be understandable, but it does present certain challenges for the reader, who may be required to consult the previous papers in order to gain a full understanding of the methods employed. It would be beneficial to expand the description of the methods (see major comment 2).

The manuscript is well-structured, but it is lengthy, and in some parts provides too much details. This makes it not easy to follow the storyline of the paper and to get the main messages.

Overall, the submitted manuscript needs careful and major revision.

**Major comments:**

(1) The entire manuscript should be streamlined to allow for a clearer storyline and clearer main messages. I recommend to shorten in particular the introduction, section 4.2/5.2, and the conclusions.

(2) Ads i previously stated, i recommend to expand the methods description, in order to allow for understanding the methods without the need to read the earlier publications. It is of particular importance to provide a detailed description of the data pre-processing employed for the calculation of the climatological background and, subsequently, the anomalies. This is because the results often depend on the manner in which the climatological background is calculated.

Since PCA in climate studies is mostly used in the S-mode (data matrix n x N with n number of stations/gridpoints and N number of timesteps and component score matrix r x N, r number of PCs) it would be beneficial to mention that here PCA is applied in P-mode (data matrix n x N with n number of parameters and N number of timesteps and component score matrix r x N, r number of PCs) (see overview table 9 in Richman,1986, Int. J. Climatology, 6, 293-335). Furthermore, I would like to ask, what is the advantage of the proposed cluster method over standard approaches like e.g. K-means clustering. Have the authors tried out such methods as well?

(3) Section 4.2/5.2: The meaning of the title "Substructure" is not apparent without reading the subsection. In summary, these subsections showing the evolution of two specific extreme winters for each cluster demonstrating sub-seasonal variability. I wonder if this information is really needed, or if accumulated information as presented in Figs. 4 and 7 is sufficient to characterize the relationship between the characteristics of the extreme winters in each cluster and synoptic-scale weather events.

Minor comments:

(1) Abstract: Should be improved. L19: "substructure" has to be explained.

(2) Introduction, L26-33: In my view, BS as hot spot of Arctic amplification should be mentioned.

(3) Introduction, L48-58: If the authors want to keep this detailed description of Arctic-midlatitude linkages, they have to be more precise in explaining tropospheric and stratospheric pathways, and the role of changes in wave activity (L54).

(4) Introduction, L69-70: "reduction in local baroclinicity following the strong sea ice retreat" In my view, this is not fully clear, since baroclinicity (e.g. expressed in term of max. Eady growth rate) is determind by vertical wind shear AND static stability.

(5) L147-151: Please explain, why you authors have used this very specific definition of BS region? How this compares to the standard definition in terms of geographical coordinates?

(6) L184: I am sorry, but I do not understand the return period of 40 years (with 50 events in an overall ensemble of 1050 simulated winters).

(7) L256: Could you provide the values of correlation between the different precursors and their changes between present day and future?

(8) L263-266: Is this the only reason for the nearly unchanged correlation between the precursor variables?

(9) L269-273: In my view, via a projection it should possible to show the PCA results for S2000 and S2100 in the same state spcae to see the changes more clearly. Did you try such an approach? Why did you decide against such an approach?

(10) Fig. 2: In my view, the inclusion of SLP anomaly plots would help for the characterization of the different clusters. Please, provide these plots as well.

(11) L405: I do not see a dipole in f_CAO*, it is strongly positive over the BS area.

(12) L581-582: "while T extremes feature patterns that favor anomalous horizontal transport of warm and cold air, respectively, towards BS."

Kind of hen-and-egg problem, better to explain it this way: patterns for T extremes are related with anomalous horizontal transport of warm and cold air, respectively, towards BS.

---

## Author Comment (AC1)

**EGUSPHERE-2024-878**

**Characteristics and dynamics of extreme winters in the Barents Sea in a changing climate**

Katharina Hartmuth, Heini Wernli, and Lukas Papritz

**Final author comments**

We thank all three reviewers for their insightful and helpful comments. We address each comment point by point below. The reviewers' comments are given in blue and our responses in black. The most important aspects of our replies and revisions are:

(1) As suggested by all reviewers, to clarify and strengthen the main messages of the study, we shorten the manuscript substantially and in particular move the case study sections 4.2 and 5.2 into a supplement.

(2) We add a more detailed explanation of the applied PCA method such that the approach can be understood without prior knowledge of earlier publications.

(3) We clarify used terminologies and methodological choices such as the concept of a seasonal "substructure", the definition of region BS based on the ensemble mean sea ice concentration and the use of different PCA projections for S2000 and S2100.

**Reviewer 1**

Overview:

This manuscript examines the changing nature of extreme winter seasons in the Barents sea region between the last 10 years of the 20th century and the last 10 years of the 21st century under RCP8.5. An extreme season is determined by the combination of 6 variables via a PCA-based method, from which 3 types of extreme seasons are identified for each time period by the clustering of seasons in a biplot. The seasonal-mean large scale anomaly patterns that contribute to these clusters are analyzed, and two example seasons are studied to understand the time evolution of daily weather systems and how they contribute. They conclude that surface boundary conditions become a less important contributor to the formation of extreme winters in the future.

General:

I think the method used here is very interesting and not one I was previously aware of, and in particular I like the idea of defining extreme seasons via a combination of factors, which to me is more intuitive for the way in which people actually experience the weather. The figures are

mostly very good and contain a lot of information. It is a very thorough breakdown of the formation of extreme seasons in the Barents Sea. However, the paper is very long in its current form and often reads like a list of results, which I think is quite common for a paper based on a PhD thesis (I know I had the same comment on a paper of mine based on my thesis!) To be publishable, I believe the authors need to really pick out the salient results and think about what the story of the paper is. I therefore recommend major revisions due to the need to clarify and rework the paper's messaging rather than much need for additional analysis.

Somewhat more specifically, the intro is quite long and I think contains some irrelevant info, such as the 3rd paragraph on linkages, while other paragraphs are repeating info already stated.

Many thanks for your positive evaluation of our study. We agree with your concerns about the paper's length and its messaging. We will substantially streamline the entire manuscript, including the introduction, hopefully leading to a more concise presentation of our main results.

The methods are a bit sparse and without reading Rothlisberger et al. 2021 and Hartmuth et al. 2022 the reader can't really understand the simulations or what the physical interpretation of the PCA method is.

In the revised manuscript we will describe the simulations as well as the PCA method in more detail such that both can be understood without the necessity to read earlier publications.

The term 'seasonal-mean anomaly' is used often in the text and I'm not certain if it means the daily mean anomaly from a seasonal average, averaged across the season, or if it means how the seasonal mean value differs from the seasonal average, so this should be made clear. Similarly, 'key variables' or 'key surface parameters' is used quite a few times but why are the chosen variables 'key'? The term is also used before the variables are ever defined (e.g. line 32). I think that 'surface parameters' on its own is probably sufficient most of the time.

With "seasonal-mean anomaly" we refer to the deviation of the seasonal mean from the ensemble mean. We aim to clarify this in the revised manuscript and use the term "seasonal anomaly" instead of "seasonal-mean anomaly".

We will further only speak of "surface parameters" instead of "key surface parameters".

Section 3 is a good overview of the statistical differences between ERA5/S2000 and S2000/S2100 but I have trouble linking this to a physical meaning. For example, what is the implication of less variance being explained by PC1 and PC2 in S2000 compared to ERA5? Why is $d_M$ less for S2100 than ERA5 if there are more seasons to choose from, also what does $d_M$ then mean physically, and does changing from 2.47 to 2.40 in the future has any physical relevance?

A lower explained variance by the first two principal components in CESM1 implies that the original 6 dimensional dataset is slightly less well represented by the first two PCs compared to ERA5. For example, the interpretation of correlations between precursor vectors is more precise

if PC1 and PC2 explain a large proportion of the overall variability (Gabriel, 1971, 1972). We will add this information in the method part of the revised manuscript.

For the analysis with ERA5 presented in HA2022, we pragmatically chose a subjective threshold of $d_M$=3 to identify extreme seasons. By using this threshold, seasons that appeared as clear outliers in the PCA biplot were defined as extreme seasons. In this study, however, we use a more systematic approach to define this threshold $d_M$ by choosing the minimum of the 50 highest $d_M$ values as the threshold (to end up with 50 extreme seasons in both S2000 and S2100). As $d_M$ is a measure for the magnitude of the combined anomalies of all six surface parameters, a slightly smaller threshold value of $d_M$ for extreme seasons in S2100 implies that a slightly smaller combined seasonal-mean anomaly is necessary to be counted as an extreme season. However, as both values only deviate by 3% we don't interpret this deviation as physically relevant.

Are the two different example seasons for each cluster necessary? It might be a way to cut back on the 'list of results' feel to the paper. I assume you probably can't present a mean time series because of how different each season's evolution of individual weather systems is, but what is added by providing two seasons?

Thank you for this suggestion to enhance the readability of our paper. We will follow your advice – see main revision aspect (1) about shortening the manuscript.

In the conclusions, paragraphs beginning at 566 and 595 both seem to just state things that are very much expected, like wet winters having a surplus of cyclones and surface boundary conditions both drive and are driven by temperature anomalies. I'm not sure these things are new results in any way, and it makes it seem like there aren't any interesting conclusions to be drawn from quite a lot of analysis. There's little here in the way of considering past work and placing results within the context of existing literature.

First, we would like to emphasize that although the link between seasonal weather system anomalies and $T/P$ anomalies is not unexpected, it is still necessary to show these results to confirm our general understanding of processes leading to seasonal extremes. In particular, the availability of weather system data for >1000 seasons in different climate states is most likely a fairly unique aspect of our study, and allows for this novel and robust statistical evaluation of the relation between weather systems and seasonal temperature/precipitation extremes. However, we agree with the reviewer that we should better emphasize the novel (and less "expected") results of our study and we will do so when revising the abstract and conclusions.

Regarding the contextualisation of our results there is indeed little existing literature investigating seasonal extremes in a statistical way, in particular from a multivariate point of view. However, we will aim to improve this part of the discussion in the revised manuscript.

Some Specific comments:

Abstract:

1: remove comma after temperatures, move to after time on line 2.

Changed as suggested.

2: predestined is a very strong word to use here, maybe something like 'anticipated to be a

Changed to "anticipated to exhibit the occurrence of surface weather extremes".

7-10 slightly confusing and long sentence

Thank you, we will aim to reformulate this sentence in a more concise way in the revised manuscript.

Introduction:

26: strongly affects -> has strongly affected

We want to emphasize that global warming is still affecting the Arctic to this date, which is why we choose present tense in this case. We checked with a native speaker who confirmed that present tense is therefore OK.

31-33 do the simulations show differing trends between models or do you mean this is in a single model?

We mean here that models show different trends for different regions and seasons. Both studies referred to investigating multiple models. However, we expect that also within a single model regionally and seasonally different trends occur. For example, we could show for CESM1 that $T$ increases much more over sea ice covered regions compared to open ocean and that this increase is much larger in winter than in summer (Hartmuth et al., 2023).

37: heat transported by the ocean?

Changed to "...further enhanced by an increase in oceanic heat transport by the Atlantic inflow".

59: remove thereof

Rephrased to "mid-latitude weather and its extremes".

64: wettening -> wetting

Changed as suggested.

Added "susceptible to sea ice loss" in line 67.

We will remove this remark about circulation uncertainty in the revised manuscript.

Deleted sentence.

Changed from "have been focusing" → "have focused".

Changed as suggested.

122-126: I'm not sure I understand what the difference between Q's 1 & 3 are, and for point 2 should probably specify that this is in a model, since it was already answered for ERA5 in your previous paper.

Q1 is solely about the surface parameters in the PCA analysis and does not involve the analysis of extreme seasons. Q3 relates to Q2 in the sense that we are interested in the change of extreme season characteristics in a warmer climate. We will clarify this and rephrase Q3 as follows:

"To what extent do the characteristics of extreme seasons in the Barents Sea change in a warmer climate?".

Changed as suggested.

Data and Methods:

143: might want to be explicit about how the surface energy budget is defined

Added the following to lines 141-142: "In addition, the sum of surface heat fluxes and surface net radiation ($H_S+H_L+R_S+R_L$) is defined as the surface energy budget and denoted by $E_S$".

Paragraph at line 146: I'm a bit confused why the Barents Sea is defined by its sea-ice cover, it seems this could be influencing your results. Does its definition then change between ensemble

members of S2000, or is it defined by an ensemble average sea ice cover? Why not just use lat/lon bounds?

We use the same region for all ensemble members in both S2000 and S2100 as it is defined by the ensemble mean sea ice concentration in S2000. We will clarify this in the revised manuscript. We do not use lon/lat bounds to avoid an overlap of our region with an area that exhibits a comparatively high sea ice concentration and where we expect to find very different characteristics of the surface parameters.

Figure 1: In terms of physical interpretation, is it important that C3 in S2100 covers two quadrants? Might want to use different colors for each cluster to make it easier to read.

In terms of physical interpretation one should rather note the position of the seasons in relation to the precursor vectors (and not in terms of the quadrants). In the case of C3 in S2100, for example, all seasons within the cluster (in both quadrants) show a positive $P$ anomaly since the pre-cursor vector for $P$ points towards the two quadrants. However, seasons in the bottom left quadrant are further characterized by a more pronounced positive $T$ anomaly, while seasons in the bottom right quadrant show a less pronounced $T$ anomaly according to the position of the $T$ vector.

174: why is the interplay between the variables largely affected by the surface type?

With "interplay" we refer to the correlations between the different variables, which are strongly modulated by the surface types. Both surface radiation as well as surface heat fluxes are a measure for interactions between the surface and the atmosphere, which substantially depend on the type of the surface, i.e., open water vs. sea ice. For example, over the open ocean, the advection of cold air will result in strongly enhanced surface heat fluxes into the atmosphere and, thus, $T$ is strongly correlated with $H_S$ and $H_L$ in winter. Over sea ice a decrease in $T$ will only result in comparatively small changes in $H_S$ and $H_L$.

In the revised manuscript we will clarify what we mean by "interactions".

Paragraph at line 196: Why define the clusters differently in S2000 and S2100, wouldn't finding the nearest 10 seasons to the S2000 cluster in the S2100 phase space be a more interesting question to examine? They're all extreme seasons, so it's not like a similar type doesn't happen in the future, we just don't see such a tight cluster (as long as I've understood correctly.) Also, since you are choosing 30/50 seasons for the clusters, it seems a bit arbitrary to claim that extremes are of a different type in the future,

First, we define clusters with the same approach in both S2000 and S2100 with the aim to obtain ten seasons that are most similar as described in lines 196ff. Furthermore, as the S2100 phase space is not identical with the S2000 phase space, it does not make sense from a physical point of view to simply pick the ten seasons that are closest to the position of the cluster in the S2000 phase space, i.e., they would not be physically the same because of the (slightly) different orientation of the vectors in the biplot.

With regard to your last remark, we are not sure which part of the text you are referring to, as we do not claim that there is a different type of extremes in the future. On the contrary, despite the clusters of S2000 and S2100 not being entirely comparable, we think that our case studies illustrate that types of extreme seasons (and the processes leading to their formation) existing in S2000 still exist in S2100 (see for example line 575f. and 622f.).

205: less -> fewer

Changed as suggested.

Interannual variability: -- Many of these sections could use more descriptive titles! Specifying this section is about a comparison, for example. Substructure was confusing to me, it's a time series analysis, etc.

Thank you for raising this important point. We will use more descriptive titles in the revised manuscript.

215-216: Why use different regions than past work, seems just to complicate the comparison.

By design, the regions are the same in both studies since we are using the same method to define the regions based on winter-mean sea ice concentration. However, because the climatological sea ice concentrations (climatological mean in ERA5 vs. ensemble mean in CESM1) differ slightly in the two datasets, the precise outlines of the regions are slightly different.

229 (& elsewhere): 'such' is often used and it's not often necessary to make the sentence clear.

We will reduce the use of "such" in the revised manuscript.

243-246: not necessary to include this info

Sentence deleted.

248 existent -> present

Changed "existent" to "present".

252: why does the increase in variability of other precursor variables give a similar decrease?

If there is an increase in the contribution of one/two variables to the overall variability, this results automatically in a decrease of the contribution of the remaining variables. To avoid confusion we modify the sentence at lines 250-252:

"The change towards more homogeneous surface conditions could explain the decreasing importance in $T$ and $H_s$ in the PCA phase space. At the same time, the increase in the contribution to the overall variability in the 6 dimensional parameter space and therefore the relative importance of the other precursor variables could similarly result in such a decrease".

Paragraph line 254: much is repetitive from previous sections

Thank you, we will shorten this section in the revised manuscript.

We assign changes in the correlation between the surface parameters largely to changes in sea ice variability. We therefore assume that the comparatively small changes of sea ice variability in BS cause the relatively small changes in correlation between surface parameters compared to other regions that experience larger changes in sea ice variability.

We added "the distribution of selected extreme season clusters in the PCA biplot" to avoid confusion. The disparity simply results from our cluster identification method but should not make a statement about the distribution of the 50 extreme seasons in both the S2000 and S2100 phase space.

Changed "which lets assume that" to "which implies that".

Thank you for this remark. We agree, however, it is hard to find a set of colors that can be applied to each panel given the various colormaps. We will think of a better color combination of these lines for the updated manuscript.

Figure 3: include a color bar for the frequency anomalies as well as bigger labels

Colorbars for the weather system frequency anomalies will be added in the updated manuscript. Labels will be slightly increased if possible following space restrictions. Note that this figure will move to the Supplement.

Changed as suggested.

369-371: confusing sentence

Splitting up the sentence for clarification: "Note, that such a frequency anomaly for a specific cluster is calculated as the deviation from climatology. This climatology is obtained as the mean over all masks that overlap with the enlarged BS region in all simulations."

397: North -> north

Changed as suggested.

Last paragraph belong in the next section

We will move the last paragraph to the beginning of section 5 in the updated manuscript.

Extreme winters in S2100

438: I think the small SST anomaly looks like a dipole or a shift in the gradient

In the BS region, the SST anomaly in cluster 1 is very small, compared to clusters 2 and 3. The small patches of anomalies are within the +/-0.1 K range, which is why we do not go into further detail regarding these anomalies.

466: sigma range-> within a standard deviation

Changed "$T$ being well below the sigma range" to "$T$ being well below one standard deviation".

489: does it become apparent from their anomalies?

It becomes apparent from the changes in values of SST and SIC as explained in the subsequent two sentences.

Discussion and Conclusion

537: I think the citation is doubled

We introduce abbreviations again for the conclusion part. Therefore, we also introduce the abbreviation for Hartmuth et al. (2022) again.

Figure 8: Really like this figure!

Thank you!

Changed as suggested.

**Reviewer 2**

This study uses a multivariate and cluster approach (considering surface air temperature, precipitation and surface energy fluxes) to identify extreme winters over the Barents Sea in the current (1990-2000) and future climate (2090-2100) from CESM large ensemble simulation. During the current and future climate, the role of atmospheric circulation (in terms of frequency of cyclones and anticyclones) and boundary surface conditions (sea ice cover and sea surface temperature) in affecting extreme winters are explored. The main conclusion is that when the sea ice edge retreats northwards in a warming climate, the boundary surface conditions play a less important role in controlling the surface variables and the extreme winters.

General comments.

The findings and methodology are valid; the analysis is thorough. Understanding the physical processes that contribute to the Arctic's extreme winters in the future climate is useful to the community. However, the paper has a lot of information and is quite long. Overall the authors should rewrite some parts (especially the Abstract and Discussion) to make the paper more readable, and better organize the main messages. I suggest a major revision.

Thank you for these remarks. Please see our main revision aspects at the beginning of the document.

Specific comments

Abstract

Line 8: surface temperature → surface air temperature

Thank you, changed as suggested.

Line 9: in a warmer climate (S2100)

Added "(S2100)".

Line 11: Substructure: This wording is new to me. Does it simply mean the synoptic (or daily) variability over a winter season?

With the term "substructure" we are referring to the daily variability in surface parameters during a winter season. For example, a particularly large positive surface air temperature could result from a prolonged period with persistent, but relatively small, above-average daily-mean surface

air temperature, or few short episodes of very strong warm air advection. This would result in strongly differing "substructures" of two such unusually warm winters.

This term has been used in several publications of the INTEXseas project about extreme seasons, such as for example in Röthlisberger et al. (2020) or Hartmuth et al. (2022). We aim to clarify this terminology in the revised manuscript (see main revision aspect (3)).

Line 14-17: I wonder if this part is important to put in the Abstract.

Thank you, we will remove this aspect from the Abstract.

Line 20: "decreasing magnitude in seasonal-mean anomalies" → I don't think this statement is correct. The magnitude of anomalies seems to be comparable in Figures 4 and 7. I do agree that the variability of the anomalies decreases, as shown in Figure 8.

The magnitude in terms of multiples of the standard deviation is indeed comparable between Figures 4 and 7, however, the absolute value of the anomalies (and thus, the variability, which is shown for example in Fig. 8) clearly decreases. To avoid confusion, we modify this formulation to "decreasing variability in seasonal-mean anomalies".

Introduction

Lines 41-43: The interannual and decadal variability is also driven by anomalous atmospheric circulation. See Siew et al. 2023 and Liu et al. 2022.

- Siew, P., Wu, Y., Ting, M., Zheng, C., Clancy, R., Kurtz, N.T. and Seager, R., 2023. Physical Links from Atmospheric Circulation Patterns to Barents–Kara Sea Ice Variability from Synoptic to Seasonal Timescales in the Cold Season. Journal of Climate, 36(22), pp.8027-8040.
- Liu, Z., Risi, C., Codron, F., Jian, Z., Wei, Z., He, X., Poulsen, C.J., Wang, Y., Chen, D., Ma, W. and Cheng, Y., 2022. Atmospheric forcing dominates winter Barents-Kara sea ice variability on interannual to decadal time scales. Proceedings of the National Academy of Sciences, 119(36), p.e2120770119.

Line 43: Should also include Siew et al. 2023 in the reference.

Thank you for mentioning these papers. We will add these references.

Line 47-59: This paragraph regarding Arctic-midlatitude teleconnection is not relevant to this study. The whole paragraph can be condensed into 1-2 sentences.

We will shorten the introduction for the revised manuscript (see main revision aspect (1)).

Lines 75-83: This paragraph has some repetitive information and it can be combined with the previous paragraph.

Thank you for this suggestion, see previous point.

Line 88-90: "we" refers to HA2022? It might be fine if the author list is the same. However, using "we" is confusing as readers might think the ERA5 analysis is done in this study. So I suggest using HA2022 to avoid confusion.

Changed as suggested.

Line 101: Add Screen 2014 in the reference.

- Screen, J.A., 2014. Arctic amplification decreases temperature variance in northern mid-to high-latitudes. Nature Climate Change, 4(7), pp.577-582.

Reference added.

Data and method:

Line 147-151: This part is unclear. Is the area of the Barents Sea fixed in the definition, or changing in different winters? Also, a bigger Barents Sea region is used for S2100 simulation, as mentioned on line 366 and Figure 5.

The area of the Barents Sea used in this study, BS, is fixed. As stated in line 147, a threshold of $SIC_{S2000}$=0.5 is applied based on the winter-mean sea ice concentration over all simulated years in S2000, i.e., the ensemble mean. We will clarify this in the updated version. In S2100, the very same region is used as mentioned in lines 149-150. The slightly enlarged area that is mentioned in line 366 is only used for the analysis of weather systems in both S2000 and S2100.

Line 140: 2-metre temperature

According to the house rules of WCD ("It is our house standard not to hyphenate modifiers containing abbreviated units") we use the term 2 m temperature.

Line 182: How many winters are there in total?

We will add the total number of winters in S2000 (1155) and S2100 (1050) at lines 137-138 in section 2.1.

Line 201-203: The two-step procedure for identifying the 3 clusters on the PCA biplot is unclear. Could you explain that more clearly in the text?

We rephrase this explanation as follows: "First, we consider only the azimuthal position of the extreme seasons in the biplot, and we determine for each group of ten adjacent extreme seasons the angle segment in the biplot that encloses this group. In a second step, the three non-overlapping groups with the smallest angle segment are chosen for the cluster analysis in Sect. 4 and 5. They are indicated by red dots in Fig. 1.

Section 3:

Section 3.1 is not very useful (what do we learn by comparing those details between the method applied on CESM and ERA5 in HA2022?). Section 3.2 can also be largely shortened.

Comparing the PCA results of both ERA5 and CESM1 data is an important step to validate the representation of the variability of surface variables in BS in the CESM1 dataset. Only if the PCA results from the CESM1 S2000 dataset are reasonably close to those from ERA5, we can continue with a comparison of climate change effects between S2000 and S2100 in the CESM1 dataset.

For the revised version of the manuscript we will shorten sections 3.1 and 3.2.

Line 215: What does KBS refer to?

The region KBS is defined in Hartmuth et al. (2022) as the part of the Kara- and Barents Seas that is mostly ice-free in the present-day climate based on the climatological sea ice edge ($SIC_{clim}$=0.5) in the ERA5 reanalysis. It largely overlaps with region BS defined in this study.

To avoid confusion, in the revised manuscript we will rewrite this sentence such that the acronym "KBS" is avoided.

Section 4.1

Figure 2: The dashed yellow line (climatological sea ice edge) is invisible over the Barents Sea region due to the overlapping with the Barents Sea box.

Due to the definition of the BS box based on the climatological sea ice edge (= dashed yellow line), both lines will overlap by design. We will adapt the figure such that the dashed yellow line is shown on top of the solid orange line for improved visibility.

Line 284: Authors should define the direction of positive energy fluxes ($E_S$).

Thank you for this important remark, we will add a sentence about this definition in section 2.1

Line 295-297: The statement related to CAOs comes from nowhere without going to Figures 3 or 4.

Thank you for this remark. This statement is independent from Figs. 3 and 4 as we link the anomalies in both $E_S$ and SIC (both shown in Fig. 2). We will rephrase the sentence to avoid the mentioning of CAOs and, thus, confusion of anticipating results shown in Figs. 3 and 4.

Section 4.2 and Sections 5.2

Line 317, 328: How these specific cases are picked (out of 10 samples ) is not mentioned.

The case studies are selected subjectively based on the evolution of the daily-mean anomalies in $T$ and $E_S$ with the goal of comparing two seasons that exhibit a similar substructure with respect to these parameters. We will mention this in the revised version. Note also that the case studies will be moved to the Supplement.

Figures 3 and 6: Overall I think the $f_c$, $f_a$ and $f_{CAO}$ analyses  (the heatmaps at the bottom of the subplots) are not very helpful. Their roles are largely accomplished by Figures 4 and 7. What do we learn here by examining their daily evolution? Also, how do these frequencies be defined on daily timescales?

The heat maps show daily-mean anomalies in cyclone frequency which are defined as the deviation of the ensemble mean on a specific day from the daily-mean climatology.

The heat map analyses of weather system frequencies are by no means represented in Figs. 4 and 7. While Figs. 4 and 7 show the seasonal-mean anomaly of $f_c$, $f_a$ and $f_{CAO}$ and, thus, only the average anomaly over a period of 90 days, the heat maps depict daily-mean anomalies and therefore allow to draw conclusions about the variability in weather system occurrence throughout the season (which is what we refer to as the substructure). For example, a positive seasonal-mean anomaly in $f_c$ could arise from heaped occurrences of cyclones only in December followed by no notable anomaly in January and February. At the same time, such an anomaly can also be the result of several large storms passing the area throughout the season. Both scenarios might cause a similar seasonal-mean anomaly in cyclone frequency but affect the other surface parameters as well as sea ice cover differently.

Line 493: sigma → standard deviation

Changed as suggested.

Section 6

Line 559: Reduced spread of $E_S$ anomalies.

The reduced spread along the x-axis denotes a reduced spread in $T$ anomalies, while the reduced spread in $E_S$ anomalies is illustrated by the lighter green and pink colors.

Line 565-590: I think these two paragraphs can be shortened and combined.

We will shorten this part of the conclusion.

**Reviewer 3**

The aim of the manuscript is to provide a comprehensive analysis of the characteristics and dynamics of extreme winters over the Barents Sea (BS) for both present-day and future conditions, based on large ensembles of climate simulations with the model CESM1. In particular, is study aims to identify the most crucial surface parameters for characterizing extreme winters over the BS, to distinguish different classes (clusters) of extreme winters over the BS and to examine their sub-seasonal evolution and the role of synoptic-scale systems (cyclones, anticyclones, cold air outbreaks (CAO) in the development of these clusters of extreme seasons. Finally, the impact of climate change on these characteristics is investigated. The authors concluded that in the future, anomalous atmospheric circulation will play a more important role than anomalous boundary conditions in the formation of extreme winters.

Given that the BS is both, a hot spot of the current Arctic climate change, exhibiting the strongest temperature amplification and sea ice retreat, and an important region for initiating processes underlying Arctic-midlatitude linkages, particularly the stratospheric pathway in winter, the topic of this study is timely and relevant.

The manuscript forms part of a series of papers on the topic of Arctic extreme seasons (Hartmuth et al., WCD 2022, Hartmuth et al., GRL, 2023). It extends those analyses with an examination of future changes of the characteristics of extreme winter seasons over the BS.

The applied methods comprise a PCA analysis and a cluster analysis, which are not commonly employed in climate studies. In my view, this combination is well suited to the aims of this study and I find it highly intriguing. Many details of the methods and the data-preprocessing can be found in the aforementioned previous papers, while this manuscript provides only a brief overview. This approach may be understandable, but it does present certain challenges for the reader, who may be required to consult the previous papers in order to gain a full understanding of the methods employed. It would be beneficial to expand the description of the methods (see major comment 2).
The manuscript is well-structured, but it is lengthy, and in some parts provides too many details. This makes it not easy to follow the storyline of the paper and to get the main messages.

Overall, the submitted manuscript needs careful and major revision.

Many thanks for the positive evaluation of our manuscript and the helpful suggestions.

Major comments:

(1) The entire manuscript should be streamlined to allow for a clearer storyline and clearer main messages. I recommend to shorten in particular the introduction, section 4.2/5.2, and the conclusions.

Thank you, see main revision aspect (1) at the beginning of the document.

(2) Ads i previously stated, i recommend to expand the methods description, in order to allow for understanding the methods without the need to read the earlier publications. It is of particular importance to provide a detailed description of the data pre-processing employed for the calculation of the climatological background and, subsequently, the anomalies. This is because the results often depend on the manner in which the climatological background is calculated.

Thank you, see main revision aspect (2).

Since PCA in climate studies is mostly used in the S-mode (data matrix n x N with n number of stations/gridpoints and N number of timesteps and component score matrix r x N, r number of PCs) it would be beneficial to mention that here PCA is applied in P-mode (data matrix n x N with n number of parameters and N number of timesteps and component score matrix r x N, r number of PCs) (see overview table 9 in Richman,1986, Int. J. Climatology, 6, 293-335). Furthermore, I would like to ask, what is the advantage of the proposed cluster method over standard approaches like e.g. K-means clustering. Have the authors tried out such methods as well?

Regarding other clustering methods, we have not tried other methods for the identification of extreme season clusters. We decided to choose a simple, yet objective method to obtain clusters that (1) contain a certain amount of seasons for a meaningful statistical interpretation and (2) contain seasons that are as similar as possible based on their (close) location in the PCA phase space. Since this approach suits our goals well, we did not explore further, more complex methods such as k-means clustering.

In the revised manuscript, we will mention the use of the P-mode and add a reference to Richman (1986).

(3) Section 4.2/5.2: The meaning of the title "Substructure" is not apparent without reading the subsection. In summary, these subsections showing the evolution of two specific extreme winters for each cluster demonstrating sub-seasonal variability. I wonder if this information is really needed, or if accumulated information as presented in Figs. 4 and 7 is sufficient to characterize the relationship between the characteristics of the extreme winters in each cluster and synoptic-scale weather events.

Thank you, see main revision aspects (1) and (3).

Minor comments:

(1) Abstract: Should be improved. L19: "substructure" has to be explained.

See main revision aspects (1) and (3).

(2) Introduction, L26-33: In my view, BS as hot spot of Arctic amplification should be mentioned.

We will add such a remark to the revised version of the manuscript.

(3) Introduction, L48-58: If the authors want to keep this detailed description of Arctic-midlatitude linkages, they have to be more precise in explaining tropospheric and stratospheric pathways, and the role of changes in wave activity (L54).

Thank you. We will shorten this part of the introduction.

(4) Introduction, L69-70: "reduction in local baroclinicity following the strong sea ice retreat" In my view, this is not fully clear, since baroclinicity (e.g. expressed in term of max. Eady growth rate) is determined by vertical wind shear AND static stability.

To avoid confusion we changed "reduction in local baroclinicity" to "reduction in horizontal temperature gradient".

(5) L147-151: Please explain why you authors have used this very specific definition of BS region? How does this compare to the standard definition in terms of geographical coordinates?

In this study, we focus on the part of the Barents Sea that is already largely ice-free in the present-day climate. Therefore, we choose the climatological sea ice edge, which is often defined by an average sea ice concentration of 50%, as the boundary of this area. Applying this method, we focus on the southern, western and central part of the Barents Sea. We aim to clarify the definition of BS compared to the standard definition of the Barents Sea in the revised manuscript.

(6) L184: I am sorry, but I do not understand the return period of 40 years (with 50 events in an overall ensemble of 1050 simulated winters).

Thank you for this remark. Indeed, 50 events within approximately 1000 years result in a return period of 20 years, not 40 years. We will adapt the paragraph in lines 182ff. as follows: "Here, the 50 winters with the largest $d_M$ are defined as extreme winters (...), which corresponds to a return period of approximately 20 years."

(7) L256: Could you provide the values of correlation between the different precursors and their changes between present day and future?

We will provide a table with the correlation values between the surface parameters in both S2000 and S2100, as well as their changes, in the supplement.

(8) L263-266: Is this the only reason for the nearly unchanged correlation between the precursor variables?

As the correlation between the chosen surface parameters depends mainly on the surface type, we assume that this is the main reason why correlation changes in this area are relatively small compared to other areas that experience substantial changes in sea ice coverage.

(9) L269-273: In my view, via a projection it should be possible to show the PCA results for S2000 and S2100 in the same state space to see the changes more clearly. Did you try such an approach? Why did you decide against such an approach?

Thank you for this thought. Yes, we tried such an approach, however, we decided against it. The main reason is that we want to analyze extreme seasons within their respective climate states. As the mean climate changes quite significantly between S2000 and S2100 we would lose a lot of information projecting the S2100 data onto the S2000 phase space. The projection of S2100 data into the S2000 phase space would mainly show a pronounced shift of the data points in the direction of the $T$ vector. However, it would not be possible to draw conclusions about the correlation between the surface variables within the future climate state as well as the unusualness of future extreme seasons relative to the future mean state.

(10) Fig. 2: In my view, the inclusion of SLP anomaly plots would help for the characterization of the different clusters. Please, provide these plots as well.

Thank you for this suggestion. We will provide the additional panels in the revised manuscript.

(11) L405: I do not see a dipole in $f_{CAO}$*, it is strongly positive over the BS area.

Thank you for pointing this out. We correct the text and replace "causing this dipole in $f_{CAO}$*" by "causing this positive anomaly in $f_{CAO}$*".

(12) L581-582: "while $T$ extremes feature patterns that favor anomalous horizontal transport of warm and cold air, respectively, towards BS."
Kind of hen-and-egg problem, better to explain it this way: patterns for $T$ extremes are related with anomalous horizontal transport of warm and cold air, respectively, towards BS.

Changed as suggested.

**References**

Gabriel, K. R.: The biplot graphic display of matrices with application to principal component analysis, Biometrika, 58, 453-467, 1971, https://doi.org/10.2307/2334381

Gabriel, K. R.: Analysis of meteorological data by means of canonical decomposition, J. Appl. Meteorol., 11, 1071-1077, 1972, https://doi.org/10.1175/1520-0450(1972)011<1071:AOMDBM>2.0.CO;2

Hartmuth, K., Boettcher, M., Wernli, H., and Papritz, L.: Identification, characteristics and dynamics of Arctic extreme seasons, Weather Clim. Dynam., 3, 89-111, 2022, https://doi.org/10.5194/wcd-3-89-2022

Hartmuth, K., Papritz, L., Boettcher, M., and Wernli, H.: Arctic seasonal variability and extremes, and the role of weather systems in a changing climate, Geophys. Res. Lett., 50, e2022GL102349, 2023, https://doi.org/10.1029/2022GL102349

Röthlisberger, M., Hermann, M., Frei, C., Lehner, F., Fischer, E. M., Knutti, R., and Wernli, H.: A new framework of identifying and investigating seasonal climate extremes, J. Clim., 34, 7761-7782, 2021, https://doi.org/10.1175/JCLI-D-20-0953.1

---

## Author Response (AR1)

**EGUSPHERE-2024-878**

**Characteristics and dynamics of extreme winters in the Barents Sea in a changing climate**

Response to the reviewers' comments by Katharina Hartmuth, Heini Wernli, and Lukas Papritz

We thank all three reviewers for their insightful and helpful comments. We address each comment point by point below. The reviewers' comments are given in blue and our responses in black. The most important aspects of our replies and revisions have been:

(1) As suggested by all reviewers, to clarify and strengthen the main messages of the study, we shortened the manuscript substantially and in particular moved the case study sections 4.2 and 5.2 into a supplement.

(2) We added a more detailed explanation of the applied PCA method such that the approach can be understood without prior knowledge of earlier publications. In addition, we extended the methods section by a more detailed description of how seasonal anomalies are defined based on a climatology.

(3) We clarified used terminologies and methodological choices such as the concept of a seasonal "substructure", the definition of region BS based on the ensemble mean sea ice concentration and the use of different PCA projections for S2000 and S2100.

Please note, that we always refer to the lines in the updated, revised manuscript (document without track changes). We supplement this document with a latexdiff-pdf showing changes since the last version of the manuscript.

**Reviewer 1**

Overview:

This manuscript examines the changing nature of extreme winter seasons in the Barents sea region between the last 10 years of the 20th century and the last 10 years of the 21st century under RCP8.5. An extreme season is determined by the combination of 6 variables via a PCA-based method, from which 3 types of extreme seasons are identified for each time period by the clustering of seasons in a biplot. The seasonal-mean large scale anomaly patterns that contribute to these clusters are analyzed, and two example seasons are studied to understand the time evolution of daily weather systems and how they contribute. They conclude that surface boundary conditions become a less important contributor to the formation of extreme winters in the future.

General:

I think the method used here is very interesting and not one I was previously aware of, and in particular I like the idea of defining extreme seasons via a combination of factors, which to me is more intuitive for the way in which people actually experience the weather. The figures are mostly very good and contain a lot of information. It is a very thorough breakdown of the formation of extreme seasons in the Barents Sea. However, the paper is very long in its current form and often reads like a list of results, which I think is quite common for a paper based on a PhD thesis (I know I had the same comment on a paper of mine based on my thesis!) To be publishable, I believe the authors need to really pick out the salient results and think about what the story of the paper is. I therefore recommend major revisions due to the need to clarify and rework the paper's messaging rather than much need for additional analysis.

Somewhat more specifically, the intro is quite long and I think contains some irrelevant info, such as the 3rd paragraph on linkages, while other paragraphs are repeating info already stated.

Many thanks for your positive evaluation of our study. We agree with your concerns about the paper's length and its messaging. We streamlined the entire manuscript, including the introduction, hopefully leading to a more concise presentation of our main results.

The methods are a bit sparse and without reading Rothlisberger et al. 2021 and Hartmuth et al. 2022 the reader can't really understand the simulations or what the physical interpretation of the PCA method is.

In the revised manuscript we added some details about the simulations as well as the PCA method to better understand both without the necessity to read earlier publications.

The term 'seasonal-mean anomaly' is used often in the text and I'm not certain if it means the daily mean anomaly from a seasonal average, averaged across the season, or if it means how the seasonal mean value differs from the seasonal average, so this should be made clear. Similarly, 'key variables' or 'key surface parameters' is used quite a few times but why are the chosen variables 'key'? The term is also used before the variables are ever defined (e.g. line 32). I think that 'surface parameters' on its own is probably sufficient most of the time.

With "seasonal-mean anomaly" we refer to the deviation of the seasonal mean from a climatology based on the temporal mean over all simulated seasons in S2000 and S2100, respectively. We clarified this in the revised manuscript (L115f.) and use the term "seasonal anomaly" instead of "seasonal-mean anomaly" to avoid confusion.

We further now only speak of "surface parameters" instead of "key surface parameters".

Section 3 is a good overview of the statistical differences between ERA5/S2000 and S2000/S2100 but I have trouble linking this to a physical meaning. For example, what is the implication of less variance being explained by PC1 and PC2 in S2000 compared to ERA5? Why is $d_M$ less for S2100 than ERA5 if there are more seasons to choose from, also what does

$d_M$ then mean physically, and does changing from 2.47 to 2.40 in the future has any physical relevance?

A lower explained variance by the first two principal components in CESM1 implies that the original six-dimensional dataset is slightly less well represented by the first two PCs compared to ERA5. For example, the interpretation of correlations between precursor vectors is more precise if PC1 and PC2 explain a larger proportion of the overall variability (Gabriel, 1971, 1972). We added this information in the method part of the revised manuscript (L168f.).

For the analysis with ERA5 presented in HA2022, we pragmatically chose a subjective threshold of $d_M$=3 to identify extreme seasons. By using this threshold, seasons that appeared as clear outliers in the PCA biplot were defined as extreme seasons. In this study, however, we use a more systematic approach to define this threshold $d_M$ by choosing the minimum of the 50 highest $d_M$ values as the threshold (to end up with 50 extreme seasons in both S2000 and S2100). As $d_M$ is a measure for the magnitude of the combined anomalies of all six surface parameters, a slightly smaller threshold value of $d_M$ for extreme seasons in S2100 implies that a slightly smaller combined seasonal-mean anomaly is necessary to be counted as an extreme season. However, as both values only deviate by 3% we don't interpret this deviation as physically relevant.

Are the two different example seasons for each cluster necessary? It might be a way to cut back on the 'list of results' feel to the paper. I assume you probably can't present a mean time series because of how different each season's evolution of individual weather systems is, but what is added by providing two seasons?

Thank you for this suggestion to enhance the readability of our paper. We followed your advice and moved the entire case study section into a supplement.

In the conclusions, paragraphs beginning at 566 and 595 both seem to just state things that are very much expected, like wet winters having a surplus of cyclones and surface boundary conditions both drive and are driven by temperature anomalies. I'm not sure these things are new results in any way, and it makes it seem like there aren't any interesting conclusions to be drawn from quite a lot of analysis. There's little here in the way of considering past work and placing results within the context of existing literature.

First, we would like to emphasize that although the link between seasonal weather system anomalies and $T/P$ anomalies is not unexpected, it is still necessary to show these results to confirm our general understanding of processes leading to seasonal extremes. In particular, the availability of weather system data for >1000 seasons in different climate states is a fairly unique aspect of our study, and allows for this novel and robust statistical evaluation of the relation between weather systems and seasonal temperature/precipitation extremes. In the revised manuscript, we clarify the relevance of such "expected" results against the background of the novel approach and unique dataset used in this study (see L418f. and 439f.).

We added some contextualisation of our results in the revised conclusions section (see L444f.). However, there is indeed little existing literature investigating seasonal extremes in a statistical way, in particular from a multivariate point of view.

Some Specific comments:

Abstract:

1: remove comma after temperatures, move to after time on line 2.

Changed as suggested.

2: predestined is a very strong word to use here, maybe something like 'anticipated to be a

Sentence deleted in the revised manuscript.

7-10 slightly confusing and long sentence

Thank you, we have split the sentence into two sentences and slightly changed their formulation to avoid confusion.

Introduction:

26: strongly affects -> has strongly affected

We want to emphasize that global warming is still affecting the Arctic to this date, which is why we choose present tense in this case. We checked with a native speaker who confirmed that present tense is therefore OK.

31-33 do the simulations show differing trends between models or do you mean this is in a single model?

We mean here that models show different trends for different regions and seasons. Both studies referred to investigated multiple models. However, we expect that also within a single model regionally and seasonally different trends occur. For example, we could show for CESM1 that $T$ increases much more over sea ice covered regions compared to open ocean and that this increase is much larger in winter than in summer (Hartmuth et al., 2023).

37: heat transported by the ocean?

Changed to "...further enhanced by an increase in oceanic heat transport by the Atlantic inflow".

59: remove thereof

Rephrased to "mid-latitude weather and its extremes".

64: wettening -> wetting

Changed as suggested.

67: susceptible to ice loss?

Sentence removed in the revised manuscript.

69: partially -> particularly

We removed this remark about circulation uncertainty in the revised manuscript.

82-83: final sentence is superfluous

Deleted sentence.

85: have been focusing -> have focused

Changed from "have been focusing" → "have focused".

101: whereby -> and

Changed as suggested.

122-126: I'm not sure I understand what the difference between Q's 1 & 3 are, and for point 2 should probably specify that this is in a model, since it was already answered for ERA5 in your previous paper.

Q1 is solely about the surface parameters in the PCA analysis and does not involve the analysis of extreme seasons. Q3 relates to Q2 in the sense that we are interested in the change of extreme season characteristics in a warmer climate. We clarified this and rephrased Q3 as follows:

"To what extent do the characteristics of extreme seasons in the Barents Sea change in a warmer climate?".

130 simulations in S2000 - > S2000 simulations

Changed as suggested.

143: might want to be explicit about how the surface energy budget is defined

Added the following to L110f.: "In addition, the sum of surface heat fluxes and surface net radiation ($H_S+H_L+R_S+R_L$) is defined as the surface energy budget and denoted by $E_S$".

Paragraph at line 146: I'm a bit confused why the Barents Sea is defined by its sea-ice cover, it seems this could be influencing your results. Does its definition then change between ensemble members of S2000, or is it defined by an ensemble average sea ice cover? Why not just use lat/lon bounds?

We use the same region for all ensemble members in both S2000 and S2100 as it is defined by the ensemble mean sea ice concentration in S2000. We clarified this in the revised manuscript and changed the sentence in L124 "...for the winter-mean sea ice concentration over all simulated years in S2000" to "...for the ensemble mean winter sea ice concentration in S2000…". We do not use lon/lat bounds to avoid an overlap of our region with an area that exhibits a comparatively high sea ice concentration and where we expect to find very different characteristics of the surface parameters.

Figure 1: In terms of physical interpretation, is it important that C3 in S2100 covers two quadrants? Might want to use different colors for each cluster to make it easier to read.

In terms of physical interpretation one should rather note the position of the seasons in relation to the precursor vectors (and not in terms of the quadrants). In the case of C3 in S2100, for example, all seasons within the cluster (in both quadrants) show a positive $P$ anomaly since the pre-cursor vector for $P$ points towards the two quadrants. However, seasons in the bottom left quadrant are further characterized by a more pronounced positive $T$ anomaly, while seasons in the bottom right quadrant, such as those in C3, show a less pronounced $T$ anomaly according to the position of the $T$ vector.

174: why is the interplay between the variables largely affected by the surface type?

With "interplay" we refer to the correlations between the different variables, which are strongly modulated by the surface types. Both surface radiation as well as surface heat fluxes are a measure for interactions between the surface and the atmosphere, which substantially depend on the type of the surface, i.e., open water vs. sea ice. For example, over the open ocean, the advection of cold air will result in strongly enhanced surface heat fluxes into the atmosphere and, thus, $T$ is strongly correlated with $H_S$ and $H_L$ in winter. Over sea ice a decrease in $T$ will only result in comparatively small changes in $H_S$ and $H_L$.

In the revised manuscript we extended L153f. by the explanation given in the first sentence above.

Paragraph at line 196: Why define the clusters differently in S2000 and S2100, wouldn't finding the nearest 10 seasons to the S2000 cluster in the S2100 phase space be a more interesting

question to examine? They're all extreme seasons, so it's not like a similar type doesn't happen in the future, we just don't see such a tight cluster (as long as I've understood correctly.) Also, since you are choosing 30/50 seasons for the clusters, it seems a bit arbitrary to claim that extremes are of a different type in the future,

First, we define clusters with the same approach in both S2000 and S2100 with the aim to obtain ten seasons that are most similar as described in L180f. Furthermore, as the S2100 phase space is not identical with the S2000 phase space, it does not make sense from a physical point of view to simply pick the ten seasons that are closest to the position of the cluster in the S2000 phase space, i.e., they would not represent the same type of extreme season because of the (slightly) different orientation of the vectors in the biplot.

With regard to your last remark, we are not sure which part of the text you are referring to, as we do not claim that there is a different type of extremes in the future. On the contrary, despite the clusters of S2000 and S2100 not being entirely comparable, we show that both specific types of extreme seasons as well as the processes leading to their formation existing in S2000 still exist in S2100.

205: less -> fewer

Changed as suggested.

Interannual variability:  -- Many of these sections could use more descriptive titles! Specifying this section is about a comparison, for example. Substructure was confusing to me, it's a time series analysis, etc.

Thank you for raising this important point. We now use more descriptive titles in the revised manuscript.

215-216: Why use different regions than past work, seems just to complicate the comparison.

By design, the regions are the same in both studies since we are using the same method to define the regions based on winter-mean sea ice concentration. However, because the climatological  sea ice concentrations (climatological mean in ERA5 vs. ensemble mean in CESM1) differ slightly in the two datasets, the precise outlines of the regions are slightly different.

229 (& elsewhere): 'such' is often used and it's not often necessary to make the sentence clear.

We reduced the use of "such" in the revised manuscript where applicable.

243-246: not necessary to include this info

Sentence deleted.

248 existent -> present

Changed "existent" to "present".

252: why does the increase in variability of other precursor variables give a similar decrease?

If there is an increase in the contribution of one/two variables to the overall variability, this results automatically in a decrease of the contribution of the remaining variables.

In the revised manuscript, we deleted this paragraph.

Paragraph line 254: much is repetitive from previous sections

Thank you, we shortened this paragraph in the revised manuscript.

We assign changes in the correlation between the surface parameters largely to changes in sea ice variability. We therefore assume that the comparatively small changes of sea ice variability in BS cause the relatively small changes in correlation between surface parameters compared to other regions that experience larger changes in sea ice variability.

We added "the distribution of selected extreme season clusters in the PCA biplot" to avoid confusion. The disparity simply results from our cluster identification method but should not make a statement about the distribution of the 50 extreme seasons in both the S2000 and S2100 phase space.

Extreme winters in S2000

Changed "which lets assume that" to "which implies that".

Thank you for this remark. We agree, however, it is hard to find a set of colors that can be applied to each panel given the various colormaps. We thus improved the visibility of the yellow and orange lines by adding an edgecolor to these contours.

Figure 3: include a color bar for the frequency anomalies as well as bigger labels

We added colorbars for the weather system frequency anomalies in the updated manuscript. Following space restrictions we slightly increased labels where possible. Note that this figure has moved to the supplement.

331: are occurring -> occur

Changed as suggested.

369-371: confusing sentence

Splitting up the sentence for clarification (L279f.): "Note that a frequency anomaly for a specific cluster is calculated as the deviation from climatology. This climatology is obtained as the mean over all masks that overlap with the enlarged BS region in all simulations."

397: North -> north

Changed as suggested.

Last paragraph belong in the next section

We moved the last paragraph to the beginning of section 5 in the updated manuscript.

Extreme winters in S2100

438: I think the small SST anomaly looks like a dipole or a shift in the gradient

In the BS region, the SST anomaly in cluster 1 is very small, compared to clusters 2 and 3. The small patches of anomalies are within the +/-0.1 K range, which is why we do not go into further detail regarding these anomalies.

466: sigma range-> within a standard deviation

Changed "$T$ being well below the sigma range" to "$T$ being well below one standard deviation". Note that this paragraph has moved to the supplement.

489: does it become apparent from their anomalies?

It becomes apparent from the changes in values of SST and SIC as explained in the subsequent two sentences.

Discussion and Conclusion

537: I think the citation is doubled

We introduce abbreviations again for the conclusion part. Therefore, we also introduce the abbreviation for Hartmuth et al. (2022) again.

Figure 8: Really like this figure!

Thank you!

557: as key region - > as a key region

Changed as suggested.

**Reviewer 2**

This study uses a multivariate and cluster approach (considering surface air temperature, precipitation and surface energy fluxes) to identify extreme winters over the Barents Sea in the current (1990-2000) and future climate (2090-2100) from CESM large ensemble simulation. During the current and future climate, the role of atmospheric circulation (in terms of frequency of cyclones and anticyclones) and boundary surface conditions (sea ice cover and sea surface temperature) in affecting extreme winters are explored. The main conclusion is that when the sea ice edge retreats northwards in a warming climate, the boundary surface conditions play a less important role in controlling the surface variables and the extreme winters.

General comments.

The findings and methodology are valid; the analysis is thorough. Understanding the physical processes that contribute to the Arctic's extreme winters in the future climate is useful to the community. However, the paper has a lot of information and is quite long. Overall the authors should rewrite some parts (especially the Abstract and Discussion) to make the paper more readable, and better organize the main messages. I suggest a major revision.

Thank you for these remarks. Please see our main revision aspects at the beginning of the document.

Specific comments

Abstract

Line 8: surface temperature → surface air temperature

Thank you, changed as suggested.

Line 9: in a warmer climate (S2100)

Added "(S2100)".

Line 11: Substructure: This wording is new to me. Does it simply mean the synoptic (or daily) variability over a winter season?

With the term "substructure" we are referring to the daily variability in surface parameters during a winter season. For example, a particularly large positive surface air temperature could result from a prolonged period with persistent, but relatively small, above-average daily-mean surface air temperature, or few short episodes of very strong warm air advection. This would result in strongly differing "substructures" of two such unusually warm winters.

This term has been used in several publications of the INTEXseas project about extreme seasons, such as for example in Röthlisberger et al. (2021) or Hartmuth et al. (2022). We clarified this terminology in the revised manuscript (see L266f).

Line 14-17: I wonder if this part is important to put in the Abstract.

Thank you, we removed this aspect from the Abstract.

Line 20: "decreasing magnitude in seasonal-mean anomalies" → I don't think this statement is correct. The magnitude of anomalies seems to be comparable in Figures 4 and 7. I do agree that the variability of the anomalies decreases, as shown in Figure 8.

The magnitude of seasonal anomalies in terms of multiples of the standard deviation is indeed comparable between S2000 and S2100, however, the absolute value of the anomalies (and thus, the variability, which is shown for example in Fig. 8 (now: Fig. 6)) clearly decreases. The mentioned sentence has been deleted in the revised manuscript.

Introduction

Lines 41-43: The interannual and decadal variability is also driven by anomalous atmospheric circulation. See Siew et al. 2023 and Liu et al. 2022.

- Siew, P., Wu, Y., Ting, M., Zheng, C., Clancy, R., Kurtz, N.T. and Seager, R., 2023. Physical Links from Atmospheric Circulation Patterns to Barents–Kara Sea Ice Variability from Synoptic to Seasonal Timescales in the Cold Season. Journal of Climate, 36(22), pp.8027-8040.
- Liu, Z., Risi, C., Codron, F., Jian, Z., Wei, Z., He, X., Poulsen, C.J., Wang, Y., Chen, D., Ma, W. and Cheng, Y., 2022. Atmospheric forcing dominates winter Barents-Kara sea ice variability on interannual to decadal time scales. Proceedings of the National Academy of Sciences, 119(36), p.e2120770119.

Line 43: Should also include Siew et al. 2023 in the reference.

Thank you for mentioning these papers. We added these references.

Line 47-59: This paragraph regarding Arctic-midlatitude teleconnection is not relevant to this study. The whole paragraph can be condensed into 1-2 sentences.

We reduced this paragraph to one sentence (L42f).

Lines 75-83: This paragraph has some repetitive information and it can be combined with the previous paragraph.

Thank you for this suggestion, we shortened this paragraph. However, as the previous paragraph deals with climate change aspects as opposed to examples of extreme weather events given in this paragraph, we did not combine both paragraphs (but instead combined this paragraph with the following one).

Line 88-90: "we" refers to HA2022? It might be fine if the author list is the same. However, using "we" is confusing as readers might think the ERA5 analysis is done in this study. So I suggest using HA2022 to avoid confusion.

Changed as suggested.

Line 101: Add Screen 2014 in the reference.

- Screen, J.A., 2014. Arctic amplification decreases temperature variance in northern mid-to high-latitudes. Nature Climate Change, 4(7), pp.577-582.

Reference added.

Data and method:

Line 147-151: This part is unclear. Is the area of the Barents Sea fixed in the definition, or changing in different winters? Also, a bigger Barents Sea region is used for S2100 simulation, as mentioned on line 366 and Figure 5.

The area of the Barents Sea used in this study, BS, is fixed. As stated in L124, a threshold of $SIC_{S2000}=0.5$ is applied based on the winter-mean sea ice concentration over all simulated years in S2000, i.e., the ensemble mean. We clarified this sentence in the updated version as mentioned in reply to the comment by reviewer 1 above. In S2100, the very same region is used as mentioned in L128. The slightly enlarged area that is mentioned in L276 is only used for the analysis of weather systems in both S2000 and S2100 (Figs. 3 and 5).

Line 140: 2-metre temperature

According to the house rules of WCD ("It is our house standard not to hyphenate modifiers containing abbreviated units") we use the term 2 m temperature.

Line 182: How many winters are there in total?

We added the total number of winters in S2000 (1155) and S2100 (1050) in L105f. in section 2.1.

Line 201-203: The two-step procedure for identifying the 3 clusters on the PCA biplot is unclear. Could you explain that more clearly in the text?

We rephrased this explanation as follows: "First, we consider only the azimuthal position of the extreme seasons in the biplot, and we determine for each group of ten adjacent extreme seasons the angle segment in the biplot that encloses this group. In a second step, the three non-overlapping groups with the smallest angle segment are chosen for the cluster analysis in Sect. 4 and 5. They are indicated by red dots in Fig. 1."

Section 3:

Section 3.1 is not very useful (what do we learn by comparing those details between the method applied on CESM and ERA5 in HA2022?). Section 3.2 can also be largely shortened.

Comparing the PCA results of both ERA5 and CESM1 data is an important step to validate the representation of the variability of surface variables in BS in the CESM1 dataset. Only if the PCA results from the CESM1 S2000 dataset are reasonably close to those from ERA5, we can continue with a comparison of climate change effects between S2000 and S2100 in the CESM1 dataset. We emphasize this more in the revised manuscript.

For the revised version of the manuscript we shortened sections 3.1 and 3.2.

Line 215: What does KBS refer to?

The region KBS is defined in Hartmuth et al. (2022) as the part of the Kara- and Barents Seas that is mostly ice-free in the present-day climate based on the climatological sea ice edge ($SIC_{clim}$=0.5) in the ERA5 reanalysis. It largely overlaps with region BS used in this study.

To avoid confusion, for the revised manuscript we rephrased this sentence and now avoid the acronym "KBS".

Section 4.1

Figure 2: The dashed yellow line (climatological sea ice edge) is invisible over the Barents Sea region due to the overlapping with the Barents Sea box.

Due to the definition of the BS box based on the climatological sea ice edge (= dashed yellow-black line), both lines will overlap by design. We adapted the figure such that the dashed yellow-black line is shown on top of the solid orange line for improved visibility.

Line 284: Authors should define the direction of positive energy fluxes ($E_S$).

Thank you for this important remark, we added a sentence about this definition in L111f.

Line 295-297: The statement related to CAOs comes from nowhere without going to Figures 3 or 4.

Thank you for this remark. This statement is independent from Figs. 3 and 4 as we link the anomalies in both $E_S$ and SIC (both shown in Fig. 2). To avoid confusion, we deleted the part of the sentence mentioning CAOs.

Section 4.2 and Sections 5.2

Line 317, 328: How these specific cases are picked (out of 10 samples) is not mentioned.

The case studies are selected subjectively based on the evolution of the daily-mean anomalies in $T$ and $E_S$ with the goal of comparing two seasons that exhibit a similar substructure with respect to these parameters which is now mentioned in the revised version. Note also that the case studies have been moved to a supplement.

Figures 3 and 6: Overall I think the $f_c$, $f_a$ and $f_{CAO}$ analyses (the heatmaps at the bottom of the subplots) are not very helpful. Their roles are largely accomplished by Figures 4 and 7. What do we learn here by examining their daily evolution? Also, how do these frequencies be defined on daily timescales?

The heat maps show daily-mean anomalies in cyclone frequency which are defined as the deviation of the ensemble mean on a specific day from the daily-mean climatology.

The heat map analyses of weather system frequencies are by no means represented in Figs. 4 and 7 (now: Figs. 3 and 5). While both figures show the seasonal-mean anomaly of $f_c$, $f_a$ and $f_{CAO}$ and, thus, only the average anomaly over a period of 90 days, the heat maps depict daily-mean anomalies and therefore allow to draw conclusions about the variability in weather system occurrence throughout the season (which is what we refer to as the substructure). For example, a positive seasonal-mean anomaly in $f_c$ could arise from the unusually frequent occurrence of cyclones only in December followed by no notable anomaly in January and February. At the same time, such a seasonal anomaly could also be the result of several large storms passing the area throughout the season. Even though both scenarios might cause a similar seasonal-mean anomaly in cyclone frequency, the effect of cyclones on the other surface parameters as well as sea ice cover is likely different.

Changed as suggested.

Line 559: Reduced spread of $E_S$ anomalies.

The reduced spread along the x-axis denotes a reduced spread in $T$ anomalies, while the reduced spread in $E_S$ anomalies is illustrated by the lighter green and pink colors.

Line 565-590: I think these two paragraphs can be shortened and combined.

We removed the first paragraph as it discussed the case study results which moved to the supplement.

**Reviewer 3**

The aim of the manuscript is to provide a comprehensive analysis of the characteristics and dynamics of extreme winters over the Barents Sea (BS) for both present-day and future conditions, based on large ensembles of climate simulations with the model CESM1. In particular, is study aims to identify the most crucial surface parameters for characterizing extreme winters over the BS, to distinguish different classes (clusters) of extreme winters over the BS and to examine their sub-seasonal evolution and the role of synoptic-scale systems (cyclones, anticyclones, cold air outbreaks (CAO) in the development of these clusters of extreme seasons. Finally, the impact of climate change on these characteristics is investigated. The authors concluded that in the future, anomalous atmospheric circulation will play a more important role than anomalous boundary conditions in the formation of extreme winters.

Given that the BS is both, a hot spot of the current Arctic climate change, exhibiting the strongest temperature amplification and sea ice retreat, and an important region for initiating processes underlying Arctic-midlatitude linkages, particularly the stratospheric pathway in winter, the topic of this study is timely and relevant.

The manuscript forms part of a series of papers on the topic of Arctic extreme seasons (Hartmuth et al., WCD 2022, Hartmuth et al., GRL, 2023). It extends those analyses with an examination of future changes of the characteristics of extreme winter seasons over the BS.

The applied methods comprise a PCA analysis and a cluster analysis, which are not commonly employed in climate studies. In my view, this combination is well suited to the aims of this study and I find it highly intriguing. Many details of the methods and the data-preprocessing can be found in the aforementioned previous papers, while this manuscript provides only a brief overview. This approach may be understandable, but it does present certain challenges for the reader, who may be required to consult the previous papers in order to gain a full understanding of the methods employed. It would be beneficial to expand the description of the methods (see major comment 2).
The manuscript is well-structured, but it is lengthy, and in some parts provides too many details. This makes it not easy to follow the storyline of the paper and to get the main messages.

Overall, the submitted manuscript needs careful and major revision.

Many thanks for the positive evaluation of our manuscript and the helpful suggestions.

Major comments:

(1) The entire manuscript should be streamlined to allow for a clearer storyline and clearer main messages. I recommend to shorten in particular the introduction, section 4.2/5.2, and the conclusions.

Thank you, see main revision aspect (1) at the beginning of the document.

(2) Ads i previously stated, i recommend to expand the methods description, in order to allow for understanding the methods without the need to read the earlier publications. It is of particular importance to provide a detailed description of the data pre-processing employed for the calculation of the climatological background and, subsequently, the anomalies. This is because the results often depend on the manner in which the climatological background is calculated.

Thank you, see main revision aspect (2).

Since PCA in climate studies is mostly used in the S-mode (data matrix n x N with n number of stations/gridpoints and N number of timesteps and component score matrix r x N, r number of PCs) it would be beneficial to mention that here PCA is applied in P-mode (data matrix n x N with n number of parameters and N number of timesteps and component score matrix r x N, r number of PCs) (see overview table 9 in Richman,1986, Int. J. Climatology, 6, 293-335). Furthermore, I would like to ask, what is the advantage of the proposed cluster method over standard approaches like e.g. K-means clustering. Have the authors tried out such methods as well?

Regarding other clustering methods, we have not tried other methods for the identification of extreme season clusters. We decided to choose a simple, yet objective method to obtain clusters that (1) contain a certain number of seasons for a meaningful statistical interpretation and (2) contain seasons that are as similar as possible based on their (close) location in the

PCA phase space. Since this approach suits our goals well, we did not explore further, more complex methods such as k-means clustering.

In the revised manuscript, we added a sentence about the use of P-mode as opposed to S-mode, referring to Richman (1986) in L150f.

(3) Section 4.2/5.2: The meaning of the title "Substructure" is not apparent without reading the subsection. In summary, these subsections showing the evolution of two specific extreme winters for each cluster demonstrating sub-seasonal variability. I wonder if this information is really needed, or if accumulated information as presented in Figs. 4 and 7 is sufficient to characterize the relationship between the characteristics of the extreme winters in each cluster and synoptic-scale weather events.

Thank you, see main revision aspects (1) and (3).

Minor comments:

(1) Abstract: Should be improved. L19: "substructure" has to be explained.

See main revision aspects (1) and (3).

(2) Introduction, L26-33: In my view, BS as hot spot of Arctic amplification should be mentioned.

We extended the following paragraph about the importance of the Barents Sea by such a remark.

(3) Introduction, L48-58: If the authors want to keep this detailed description of Arctic-midlatitude linkages, they have to be more precise in explaining tropospheric and stratospheric pathways, and the role of changes in wave activity (L54).

Thank you. We shortened this part of the introduction (see comment by reviewer 2 above).

(4) Introduction, L69-70: "reduction in local baroclinicity following the strong sea ice retreat" In my view, this is not fully clear, since baroclinicity (e.g. expressed in terms of max. Eady growth rate) is determined by vertical wind shear AND static stability.

Sentence removed.

(5) L147-151: Please explain why you authors have used this very specific definition of BS region? How does this compare to the standard definition in terms of geographical coordinates?

In this study, we focus on the part of the Barents Sea that is already largely ice-free in the present-day climate. Therefore, we choose the climatological sea ice edge, which is often

defined by an average sea ice concentration of 50%, as the boundary of this area. Applying this method, we focus on the southern, western and central part of the Barents Sea. We added a sentence in L125f. to further clarify our definition of BS and its comparison to the standard definition of the Barents Sea in the revised manuscript.

(6) L184: I am sorry, but I do not understand the return period of 40 years (with 50 events in an overall ensemble of 1050 simulated winters).

Thank you for this remark. Indeed, 50 events within approximately 1000 years result in a return period of 20 years, not 40 years. We adapted the paragraph in L161f. as follows: "Here, the 50 winters with the largest $d_M$ are defined as extreme winters (...), which corresponds to a return period of approximately 20 years."

(7) L256: Could you provide the values of correlation between the different precursors and their changes between present day and future?

We added a table with the correlation values between the surface parameters in both S2000 and S2100, as well as their changes, to the supplement.

(8) L263-266: Is this the only reason for the nearly unchanged correlation between the precursor variables?

As the correlation between the chosen surface parameters depends mainly on the surface type, we assume that this is the main reason why correlation changes in this area are relatively small compared to other areas that experience substantial changes in sea ice coverage.

(9) L269-273: In my view, via a projection it should be possible to show the PCA results for S2000 and S2100 in the same state space to see the changes more clearly. Did you try such an approach? Why did you decide against such an approach?

Thank you for this thought. Yes, we tried such an approach, however, we decided against it. The main reason is that our goal is to analyze extreme seasons within their respective climate states. As the mean climate changes quite significantly between S2000 and S2100 we would lose a lot of information projecting the S2100 data onto the S2000 phase space. The projection of S2100 data into the S2000 phase space would mainly show a pronounced shift of the data points in the direction of the $T$ vector. However, it would not be possible to draw conclusions about the correlation between the surface variables within the future climate state as well as the unusualness of future extreme seasons relative to the future mean state.

(10) Fig. 2: In my view, the inclusion of SLP anomaly plots would help for the characterization of the different clusters. Please, provide these plots as well.

Thank you for this suggestion. We extended Figs. 2 and 5 (now: Figs. 2 and 4) by additional panels in the revised manuscript.

(11) L405: I do not see a dipole in $f_{CAO}$*, it is strongly positive over the BS area.

Thank you for pointing this out. We corrected the text and replaced "causing this dipole in $f_{CAO}$*" by "causing this positive anomaly in $f_{CAO}$*".

(12) L581-582: "while $T$ extremes feature patterns that favor anomalous horizontal transport of warm and cold air, respectively, towards BS."
Kind of hen-and-egg problem, better to explain it this way: patterns for $T$ extremes are related with anomalous horizontal transport of warm and cold air, respectively, towards BS.

Changed as suggested.

**References**

Gabriel, K. R.: The biplot graphic display of matrices with application to principal component analysis, Biometrika, 58, 453-467, 1971, https://doi.org/10.2307/2334381

Gabriel, K. R.: Analysis of meteorological data by means of canonical decomposition, J. Appl. Meteorol., 11, 1071-1077, 1972,
https://doi.org/10.1175/1520-0450(1972)011<1071:AOMDBM>2.0.CO;2

Hartmuth, K., Boettcher, M., Wernli, H., and Papritz, L.: Identification, characteristics and dynamics of Arctic extreme seasons, Weather Clim. Dynam., 3, 89-111, 2022,
https://doi.org/10.5194/wcd-3-89-2022

Hartmuth, K., Papritz, L., Boettcher, M., and Wernli, H.: Arctic seasonal variability and extremes, and the role of weather systems in a changing climate, Geophys. Res. Lett., 50,
e2022GL102349, 2023, https://doi.org/10.1029/2022GL102349

Röthlisberger, M., Hermann, M., Frei, C., Lehner, F., Fischer, E. M., Knutti, R., and Wernli, H.: A new framework of identifying and investigating seasonal climate extremes, J. Clim., 34,
7761-7782, 2021, https://doi.org/10.1175/JCLI-D-20-0953.1

---

## Author Response (AR2)

**EGUSPHERE-2024-878**

**Characteristics and dynamics of extreme winters in the Barents Sea in a changing climate**

Response to the reviewers' comments by Katharina Hartmuth, Heini Wernli, and Lukas Papritz

We thank all reviewers again for their helpful comments. We address each comment point by point below. The reviewers' comments are given in blue and our responses in black.

Please note, that we always refer to the lines in the updated, revised manuscript (document without track changes). We supplement this document with a latexdiff-pdf showing changes since the last version of the manuscript.

**Reviewer 1**

Overall, the manuscript has been improved from its previous version and the authors have taken suggestions, such as streamlining the narrative and shortening some sections, and generated a better article. It is clear that a lot of work has gone into this iteration of the work and I have only a few very minor comments, mainly technical, after which I recommend publication.

Minor:

L24: Aren't trends in summer and autumn sea ice expected to be largest?

Thank you for this remark. We realized there is some repetition here anyways, which is why we shortened and combined this sentence with the previous one:
"Global climate models project continuing large changes in Arctic sea ice extent and surface conditions in the coming century (...), with the prospect of an ice-free Arctic during September within a few decades (...)."

L150: What are P mode and S mode and why use one rather than the other?

We added this remark about the used PCA mode following the comment of reviewer 3 (please see comment and our reply). Following Richman (1986), the PCA mode differs depending on the definition of the component score matrix and, more precisely, which parameters are chosen as variables and fixed entity. In our case, we combine a variety of parameters for a fixed area which refers to P-mode as opposed to S-mode, where one would combine a number of different gridpoints/regions for a fixed field. That we use P-mode in our study follows by design of our method that is based on the combination of multiple surface parameters in a fixed region.

L194: It might be a good idea to add a few words here on potential issues with precip in ERA5, perhaps particularly in a remote region like the Barents Sea, which would be less constrained by observations, which could be biasing the ERA5 dependence on precip for extremes.

We added the following sentence in L194: "Note however that due to the remoteness of the study area the accuracy of ERA5 precipitation fields is potentially limited."

L282-284: I do not understand what issue is being pointed out here. The conclusions are repetitive, this section could still be somewhat shortened.

Here we point out that the amount of values contributing to the climatology (that is used to calculate seasonal anomalies in weather system frequencies) is not the same for each grid point and, in particular, decreases with increasing distance to BS, as the climatology is obtained as the mean of all cyclone/anticyclone/CAO masks overlapping with the enlarged BS region. This is important to consider when interpreting the results as an anomaly that occurs at a large distance to BS could be based on a less "robust" climatology compared to an anomaly within BS.

Technical:

L1: a trend in a decline would be something like the second derivative of sea ice area with respect to time, when I think the authors mean to imply large negative trends or large declines rather than large trends in the decline.

Changed to "...is experiencing large declines in sea ice and increasing surface temperatures…".

L2 remove comma after time.

Changed as suggested.

L43: still discussed, it has been shown -> a topic of debate/it has been argued

Changed as suggested.

L81 - 82: add comma after Further and extremes, and -> as well as

Changed as suggested.

L89: To make this sentence clearer, I'd recommend inverting it: In a warmer climate, is there a change in the relative importance….

Changed as suggested.

L278: remove comma after winter, if -> whether

Changed as suggested.

L298 (& L375, possibly elsewhere): favors -> favours

As we consistently use American English throughout the entire manuscript, we will keep "favor" and not change it towards "favour".

L355 correlate -> correlates

Changed as suggested.

L388: remove comma after indicates

Changed as suggested.

L463: 'and by the surface' -> and drive the atmosphere?

Changed sentence to "can be either driven by the atmosphere or driving the atmosphere."

L477-479: run-on-sentence

Thank you for this remark, subdivided into two sentences in the updated manuscript.

**Reviewer 3**

Thanks for the authors' effort in addressing my questions and shortening the manuscript by putting some materials in the supplementary. I recommend publication in WCD subject to minor revisions.

The substructure (L266-271 for S2000 and L366-370 for S2100) should be discussed after the synoptic weather systems are introduced (Figures 3 and 5 respectively)

Thank you for this suggestion. However, we think that mentioning the substructure before the respective sections is an important motivation for the analysis of the weather system anomalies (see L269-273). Further, there is no mention of weather systems at all in L366-370 (only surface parameters which are discussed in exactly this section) and, thus, moving this paragraph to the end of section 5.2 would result in confusion.

L409: "which combined yield" → "which combinedly yield"

Changed as suggested.

L600: "resulting in comparatively small changes in sea ice variability in a warmer climate" → Does this contradict the sea ice edge retreat in S2100?

No, this statement does not contradict the sea ice edge retreat in S2100. What we want to emphasize here is that in this study we are analyzing a sub-region in the Arctic which - despite the overall strong sea ice retreat - is already mostly ice-free in the present-day climate. As we show in the study, it is still strongly affected by the sea ice retreat, particularly in its surrounding regions. However, as the region itself shows a comparatively small change in its sea ice cover, changes in the variables governing atmospheric variability in a warmer climate are relatively small compared to regions that experience a much larger loss in sea ice cover.

L239: It is good to mention again the direction of surface energy fluxes

Added "a positive ES* (i.e., net energy flux into the surface)".

**References**

Richman, M. B.: Rotation of principal components, Int. J. Climatol., 6, 293-335, 1986, https://doi.org/10.1002/joc.3370060305